# Unveiling the spatially confined oxidation processes in reactive electrochemical membranes

Yuyang Kang [1,2], Zhenao Gu [1,2,3] ✉, Baiwen Ma[1,2,4], Wei Zhang[1,5], Jingqiu Sun[1,2], Xiaoyang Huang [4], Chengzhi Hu [1,2,3], Wonyong Choi [4] & Jiuhui Qu [1,2] ✉

Electrocatalytic oxidation offers opportunities for sustainable environmental remediation, but it is often hampered by the slow mass transfer and short lives of electro-generated radicals. Here, we achieve a four times higher kinetic constant (18.9 min$^{-1}$) for the oxidation of 4-chlorophenol on the reactive electrochemical membrane by reducing the pore size from 105 to 7 μm, with the predominate mechanism shifting from hydroxyl radical oxidation to direct electron transfer. More interestingly, such an enhancement effect is largely dependent on the molecular structure and its sensitivity to the direct electron transfer process. The spatial distributions of reactant and hydroxyl radicals are visualized via multiphysics simulation, revealing the compressed diffusion layer and restricted hydroxyl radical generation in the microchannels. This study demonstrates that both the reaction kinetics and the electron transfer pathway can be effectively regulated by the spatial confinement effect, which sheds light on the design of cost-effective electrochemical platforms for water purification and chemical synthesis.

Electrocatalysis has been regarded as a promising approach for sustainable environmental remediation[1,2] as well as chemical synthesis[3-5]. Although significant progress has been made in developing a variety of active electrocatalysts, harnessing the full power of electrocatalysis is usually hampered by the slow mass transport process[6-8]. To address this problem, reactive electrochemical membranes (REMs), or flow-through electrodes, have been recently developed as a promising solution (Supplementary Fig. 1a)[9-11]. Owing to the multiscale pore networks in the electrode, the electrochemical reactions are spatially confined in the microchannels of REMs[12,13]. In this way, the diffusion layer thickness of REMs can be greatly reduced compared to that of conventional plate electrodes, contributing to enhanced mass transport of reactants[14,15]. The utilization efficiency of active sites can also be optimized as they are fully available to the reactant molecules in the microchannels.

Although REMs have been widely employed in electrochemical systems including water purification reactors[16,17], electrochemical synthesis cells[18,19], and redox flow batteries[7], their design principle for specific applications is unclear due to the lack of structure-performance relationships[7,20]. The gap arises from the complex integration of mass transfer and electrochemical reactions in the spatially confined pores. One of the most crucial factors determining the performance of REMs is the pore diameter, which directly relates to the mass transfer[21,22] and electrochemical properties[8,23]. However, reported REMs differ from each other in terms of catalysts and pore geometry, making it almost impossible to quantitatively assess the impact

[1]State Key Laboratory of Environmental Aquatic Chemistry, Research Center for Eco-Environmental Sciences, Chinese Academy of Sciences, Beijing 100085, China. [2]University of Chinese Academy of Sciences, Beijing 100049, China. [3]National Engineering Research Center of Industrial Wastewater Detoxication and Resource Recovery, Beijing 100085, China. [4]KENTECH Institute for Environmental & Climate Technology, Korea Institute of Energy Technology (KENTECH), Naju 58330, Korea. [5]Center for Water and Ecology, State Key Joint Laboratory of Environment Simulation and Pollution Control, School of Environment, Tsinghua University, Beijing 100084, China. ✉e-mail: zagu@rcees.ac.cn; jhqu@rcees.ac.cn

of pore diameter. Besides, it is challenging to elucidate whether and how the reaction mechanism would change with the pore diameter (e.g., direct electron transfer (DET) and indirect electron transfer mediated by radicals[24,25]). This lack of understanding of the spatially confined electrochemical reactions precludes the development of high-efficiency REMs and electrochemical devices.

In this study, reaction processes in microchannels are investigated by conducting experimental studies and finite element simulations. The mass transport and electrochemical oxidation processes in REMs are quantitatively investigated. A theoretical model is constructed to elucidate the distribution of the potential and current in the microchannels, where the depletion of reactants and hydroxyl radicals (•OH) near the electrode surface are visually observed. Overall, this study unveils the spatial confinement effect in REMs for regulating the reaction mechanism and emphasizes the significance of overcoming the mass transport limitation prevalent in electrochemical systems.

## Results

### Characterization of REMs

The electrochemical reactions in REMs are illustrated in Fig. 1a. In electrochemical oxidation processes, the main reaction pathways include DET on the electrode surface and indirect electron transfer mediated by electro-generated radicals[24,25]. Considering the extremely short lifetimes of radicals (e.g., $10^{-9}$–$10^{-6}$ s for hydroxyl radical, •OH[26,27]), the electrocatalytic oxidation process usually occurs in a thin layer near the electrode surface[28,29], leading to the depletion of reactants from the bulk solution. According to Fick's law, the mass transport rate is inversely proportional to the thickness of the depletion layer. The diffusion length can be considerably reduced by decreasing the pore size of the REM, thus contributing to accelerated mass transport. Meanwhile, the potential and current can change drastically with the depth in confined channels[30,31], which would cause large uncertainty in the electrochemical reaction process. Consequently, a comprehensive discussion of the spatial confinement effect on the reaction kinetics and mechanism is necessary.

To quantitively assess the spatial confinement effect, we constructed a series of REMs using Ti membranes as the conductive substrates (Supplementary Fig. 2). The interconnected pore network served as water channels, on which defective $TiO_2$ nanosheets were fabricated as electrocatalysts (Supplementary Fig. 3). The REMs were denoted by their predominant pore sizes, as $REM_{7\,\mu m}$, $REM_{17\,\mu m}$, $REM_{45\,\mu m}$, and $REM_{105\,\mu m}$, respectively (Fig. 1b). Over 45% of the total pore volume falls within the ±40% range around the predominant size, indicating that the selected pore size is representative (Supplementary Fig. 4). It should be noted that the Brunauer−Emmett−Teller (BET) surface area of REMs increased with the reduction of pore size (Supplementary Table 1). For instance, the surface area of $REM_{7\,\mu m}$ was determined to be 10.85 $m^2\,g^{-1}$, which is four times that of $REM_{105\,\mu m}$. Such a trend in BET surface area is consistent well with the electrochemical double-layer capacitance (Supplementary Fig. 5) and mercury intrusion results (Supplementary Table 1).

Meanwhile, pore size showed little difference in the crystalline structure of REMs, as revealed using X-ray diffraction (XRD, Supplementary Fig. 6). Following annealing in an argon atmosphere, the color of REMs changed to a grayish black, indicating the existence of crystal defects (Supplementary Fig. 7). The electronic structure of REMs was characterized using electron spin resonance (ESR) spectroscopy and X-ray photoelectron spectroscopy (XPS). A typical sign of oxygen vacancies[32] at $g = 2.003$ was observed (Supplementary Fig. 8), indicating the presence of defects in the $TiO_2$ nanosheets. After annealing in an argon atmosphere, the oxygen vacancy increased, as shown in the O $1s$ XPS band (Supplementary Fig. 9a, b)[33]. Additionally, the Ti $2p$ peak and the valence band edge shifted toward lower binding energy (Supplementary Fig. 9c, d), indicating an increased electron density with the introduction of oxygen vacancy, which substantially improves the conductivity and electrochemical activity[34,35].

The REMs with different pore diameters serve as an ideal platform to investigate the spatial confinement effect of microchannels. In the voltammetry curve in Fig. 1c, the REMs show a relatively high onset potential of 2.1 $V_{RHE}$ for oxygen evolution, which provides an

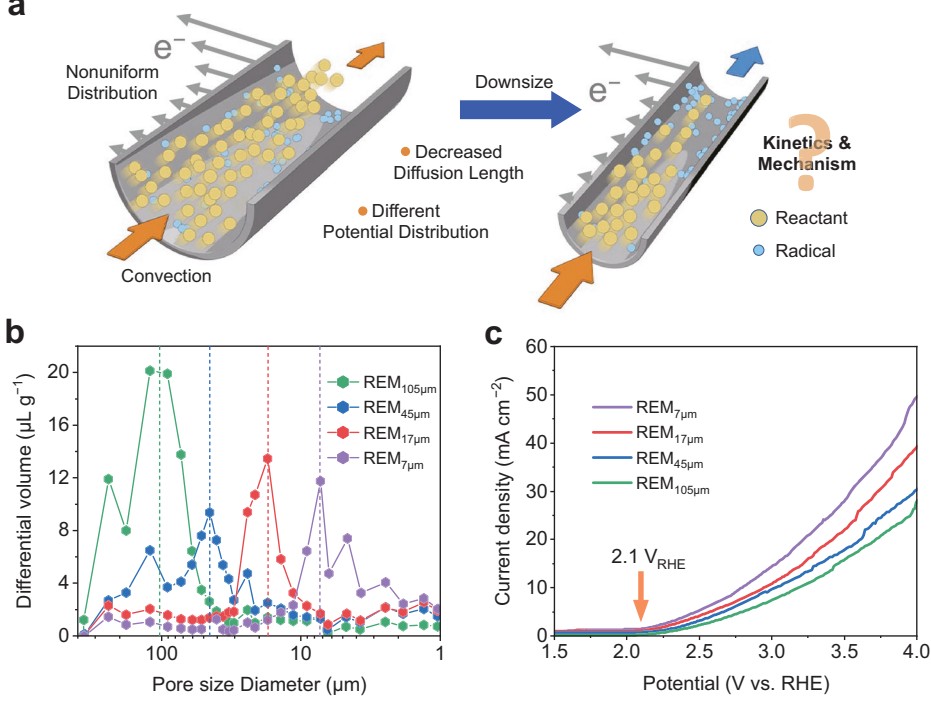

**Fig. 1 | Schematic, structure characterization, and electrochemical properties of reactive electrochemical membranes (REMs). a** Schematic illustration of microchannels with different diameters depicting the spatial confinement effect in REMs. **b** Differential mercury intrusion of REMs. **c** Linear sweep voltammetry curves of different REMs, where a similar onset potential of 2.1 $V_{RHE}$ can be observed for all four REMs. Electrolyte: 0.33 M NaClO₄.

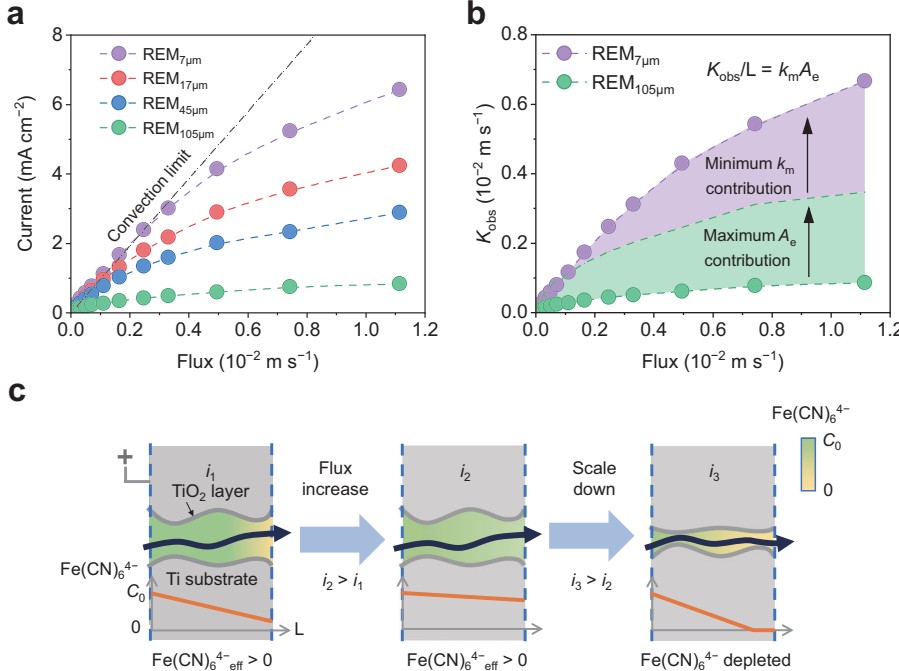

**Fig. 2 | Mass transfer in REMs. a, b** Oxidation current of $Fe(CN)_6^{4-}$ (**a**) on REMs at different fluxes and the corresponding $K_{obs}$ (**b**) of $REM_{7\,\mu m}$ and $REM_{105\,\mu m}$. **c** Schematic diagram of the oxidation of $Fe(CN)_6^{4-}$ on REMs at different fluxes. The convection limit shown in **a** represents the situation where all the influent $Fe(CN)_6^{4-}$ ions are oxidized. Highlighted areas in **b** correspond to the respective contributions of specific surface area and pore diffusion. Experiments were conducted at 1.91 $V_{RHE}$ in an electrolyte containing 0.1 mM $Fe(CN)_6^{4-}$, 0.2 mM $Fe(CN)_6^{3-}$, and 0.33 M $NaClO_4$.

advantage for the generation of •OH from water dissociation in competition with the oxygen evolution reaction[36]. Such an electrochemical platform gives us an opportunity to simultaneously investigate the direct and indirect electron transfer processes in microchannels. The current increased as the pore size decreased, following a similar trend observed for the specific surface area of REMs. However, according to the Butler−Volmer equation[37], the current is expected to be proportional to the electrode area, which is not completely consistent with the observed result. This discrepancy may be attributed to the nonuniform potential distribution within the pores, which is discussed in detail below. The Nyquist curves of REMs exhibited low series resistance of ca. 4.5 Ω, indicating the high conductivity of Ti substrate (Supplementary Fig. 10)[38]. Additionally, $REM_{7\,\mu m}$ exhibited the smallest arc radius, indicating the fastest electron transfer across the electrode-solution interface. Moreover, steady potentials and electrooxidation performance were maintained throughout the experimental duration at a fixed current density, underscoring the capability of REMs to serve as model electrodes (Supplementary Fig. 11).

**Mass transfer in spatially confined REMs**
To gain insights into the mass transfer mechanism of reactants toward the electrode surface, the oxidation current of $Fe(CN)_6^{4-}$, a model reactant with high intrinsic electrochemical reactivity, was measured under different fluxes[39]. To exclude the impact of charge transfer limitation, a relatively low concentration of 0.1 mM $Fe(CN)_6^{4-}$ was employed for the electrochemical reaction (Supplementary Figs. 12 and 13)[40]. At relatively low membrane flux (e.g., flux $<0.3 \times 10^{-2}$ m s$^{-1}$), the current on $REM_{7\,\mu m}$ linearly changed with flux and was nearly identical to the convection limit[39] (Fig. 2a). This indicates that almost all the $Fe(CN)_6^{4-}$ ions that traverse the membrane were oxidized and the reaction was limited by the convection process[39] (Fig. 2c). In contrast, the current observed on $REM_{105\,\mu m}$ was significantly lower than

that on $REM_{7\,\mu m}$, indicating the slower mass transfer of $Fe(CN)_6^{4-}$ within relatively large channels (Fig. 2c). As the flux increased, the anodic current gradually increased, indicating the alleviated concentration polarization of $Fe(CN)_6^{4-}$ (Fig. 2a, c). As the flux continuously increased, the current on $REM_{7\,\mu m}$ gradually deviated from the linear region, suggesting that the reaction is also controlled by the diffusion processes of the reactant molecules[41].

To quantitatively investigate the mass transfer process, the observed mass transfer rate ($K_{obs}$) was determined from the anodic current according to Eq. (1)[39] (Fig. 2b). The mass transfer rate can also be expressed as the product of the mass transfer coefficient ($k_m$) and surface area ($A_e$), which is widely employed and shown in Eq. (2)[42]. However, in the REM system, the increase in surface area does not result in a proportional improvement in kinetics, as validated by the current curves in Fig. 1c. Two factors can contribute to this. First, the potential distribution within the microchannels is nonuniform, influencing the effective reaction area, and it is discussed in detail below. Second, there is a strong interaction between mass transfer and effective surface area, where one factor substantially affects the other. For instance, the influent $Fe(CN)_6^{4-}$ ions in $REM_{7\,\mu m}$ can be completely oxidized at a relatively low flux, whereas in the $REM_{105\,\mu m}$ at the identical flux, they cannot be fully oxidized (Fig. 2c). Therefore, the effective surface area of $REM_{7\,\mu m}$ decreases.

Although quantifying the contribution of surface area and mass transfer is challenging, we can roughly categorize the kinetics enhancement into two parts: (i) the maximum $A_e$ contribution (i.e., fourfold) and (ii) the minimum $k_m$ contribution (Fig. 2b). At fluxes $>0.5 \times 10^{-2}$ m s$^{-1}$, $REM_{7\,\mu m}$ exhibited ~sevenfold higher $K_{obs}$ than $REM_{105\,\mu m}$. Considering that the surface area of $REM_{7\,\mu m}$ is only fourfold greater than that of $REM_{105\,\mu m}$ (Supplementary Table 1), the enhanced $K_{obs}$ value of $REM_{7\,\mu m}$ cannot be merely attributed to the relatively large surface area. The accelerated mass diffusion

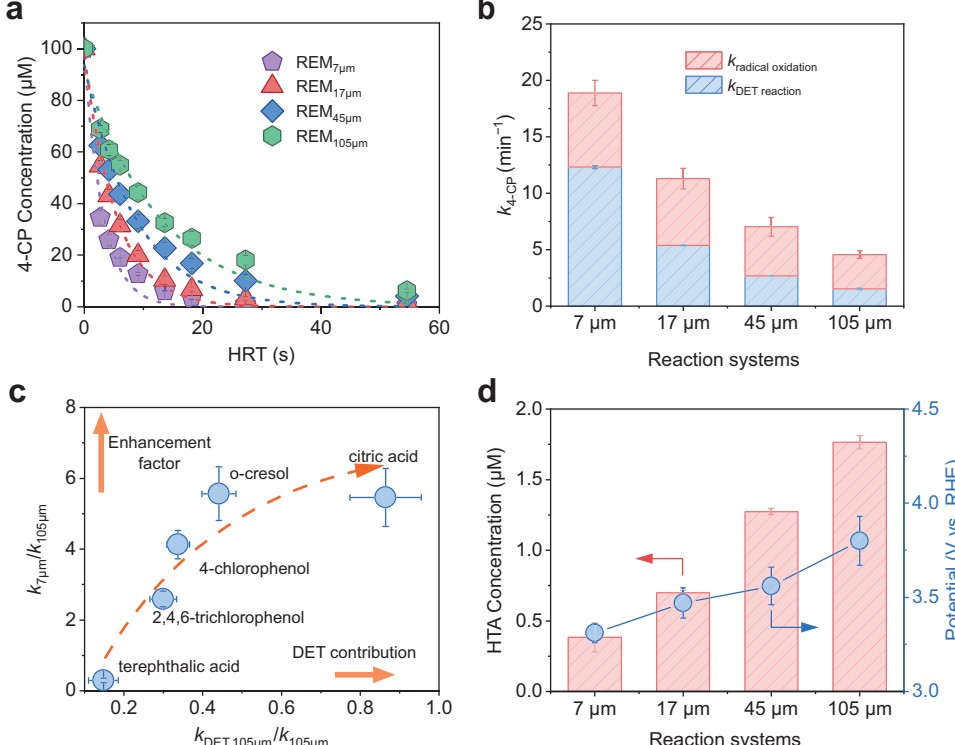

**Fig. 3 | Performance of electrochemical oxidation of organics on REMs.**
**a**, **b** Effluent 4-chlorophenol (4-CP) concentration (**a**) as a function of HRT in different REMs. **b** The corresponding pseudo-first-order kinetic constants of 4-CP degradation. **c** Relationship between enhancement factors and the contribution of direct electron transfer (DET) in $REM_{105\,\mu m}$ for different model reactants.

**d** Production of 2-hydroxyterephthalic acid (HTA) at a hydraulic residence time (HRT) of 2.7 s (left axis) and the electrode potential in degradation experiment (right axis) on REMs. The initial concentration of organics: 100 µM. Current: 19.7 mA cm$^{-2}$. Reaction area: 2.54 cm$^2$. pH = 7. Error bars represent the data from duplicate tests.

contributed by smaller pores is another important reason. Supplementary Fig. 14 depicts the relationship between $k_m A_e$ and flux for various REMs, where the electrode performance from the literature was also presented for comparison. $REM_{7\,\mu m}$ is among the best-performing electrodes with high mass transfer rate and reaction kinetics. This enables rapid contaminant removal and chemical transformation.

**Electrochemical oxidation kinetics on REMs**

The electrochemical performance of REMs was evaluated by using 4-chlorophenol (4-CP) as a model organic substrate, which is a representative environmental pollutant[43]. With the increase in hydraulic residence time (HRT), the residual 4-CP concentration in the effluent gradually decreased (Fig. 3a). Under a fixed current density of 19.7 mA cm$^{-2}$, 94.0% degradation of 100 µM 4-CP was achieved within just 13.6 s on $REM_{7\,\mu m}$, while only 67.4% of 4-CP was degraded on $REM_{105\,\mu m}$, demonstrating the superiority of the REM with small pores. No significant pH change was observed throughout the course of the experiment. The corresponding pseudo-first-order kinetic constants of different REMs were then calculated to quantify the spatial confinement effect for 4-CP degradation. As shown in Fig. 3b, the degradation kinetic constant is greatly improved with the decrease in pore size. Specifically, $REM_{7\,\mu m}$ exhibits a kinetic constant as high as 18.9 min$^{-1}$, which is 4.1 times larger than that of $REM_{105\,\mu m}$ (4.6 min$^{-1}$). The 4-CP degradation experiments at current densities of 9.8 mA cm$^{-2}$ and 39.3 mA cm$^{-2}$ were also conducted (Supplementary Fig. 15). The degradation performance of REMs was significantly enhanced, especially on $REM_{7\,\mu m}$, with an increase in the current density. The kinetic constant on $REM_{7\,\mu m}$ (36.3 min$^{-1}$) was 5.2 times that of $REM_{105\,\mu m}$ (7.1 min$^{-1}$) at 39.3 mA cm$^{-2}$. This can be primarily attributed to the relatively fast mass transfer on $REM_{7\,\mu m}$, which well matched the

elevated current. Moreover, the potential increase at a higher current could facilitate $REM_{7\,\mu m}$ in generating more •OH, which is discussed in the next section. The degradation performance of 4-CP on $REM_{7\,\mu m}$ under room light and dark conditions did not differ (Supplementary Fig. 16). This further excludes the influence of light on the electro-oxidation reactions.

To further assess the application potential of REMs in pollutant mineralization, the total organic carbon (TOC) values were measured. At an HRT of 54.5 s, the TOC removal rate on $REM_{7\,\mu m}$ was 96%, whereas only 64% of TOC was removed on $REM_{105\,\mu m}$ (Supplementary Fig. 17a). The corresponding mineralization current efficiency was then calculated based on Supplementary equation (1)[44] (Supplementary Fig. 17b). $REM_{7\,\mu m}$ can achieve a current efficiency of 56% at an HRT of 6.1 s, which is three times that of $REM_{105\,\mu m}$ (19%). The energy consumption was then calculated, where both the electrical and pumping energy were considered (Supplementary Fig. 18, Supplementary Table 2). Although $REM_{7\,\mu m}$ requires higher pumping energy due to the higher transmembrane pressure, the total energy consumption to achieve 90% 4-CP removal is 0.86 kWh per m$^3$ wastewater, which is 29% of $REM_{105\,\mu m}$ (2.95 kWh m$^{-3}$).

In general, electrochemical oxidation of organics may proceed via DET reaction on the electrode surface and indirect electron transfer mediated by radicals. The radicals involved in the degradation process were investigated via ESR with 5,5-dimethyl-1-pyrroline N-oxide (DMPO) serving as the trapping agent. A distinct ESR signal of DMPO-•OH with a typical peak intensity ratio of 1:2:2:1 was observed[45], indicating the generation of •OH on REMs (Supplementary Fig. 19). Further, the signal of DMPO-•OH is lower on REMs with smaller pore sizes. This result contrasts with the fact that $REM_{7\,\mu m}$ exhibited optimal electrooxidation performance (Fig. 3a). This discrepancy implies the presence of distinct reaction mechanisms in REMs with different pore

sizes. To distinguish the contribution of radical oxidation on 4-CP degradation, *tert*-butanol (TBA), a well-recognized •OH scavenger, was added to the reaction system[46]. Although the 4-CP degradation kinetics can be significantly reduced in the presence of 10 mM TBA, further increasing TBA concentration can enhance the quenching effect to some extent (Supplementary Fig. 20). Ultimately, 200 mM TBA was utilized to ensure complete quenching. In the presence of 200 mM TBA, the degradation process on different REMs was inhibited at varying degrees (Fig. 3b and Supplementary Fig. 21). The kinetic constant slightly dropped to 12.3 min$^{-1}$ on REM$_{7\,\mu m}$, indicating that radical oxidation accounted for 34.9% of the degradation capability. In contrast, radical oxidation was responsible for 66.3% of 4-CP degradation on REM$_{105\,\mu m}$. The DET reaction accounted for a larger percentage of the REM with a smaller pore size. From an industrial application point of view, such a spatial confinement effect of REM can be crucial for selective electrosynthesis and pollutant degradation[47,48].

To further investigate the spatial confinement effect on the reaction mechanism, the oxidation experiments of various organics, including terephthalic acid (TA), 2,4,6-trichlorophenol, *o*-cresol, and citric acid, were conducted (Supplementary Fig. 22). The enhancement factors of the kinetic constant by reducing the pore size from 105 μm to 7 μm ($k_{7\,\mu m}/k_{105\,\mu m}$) were obtained (Fig. 3c). The contribution of the DET route on the reaction kinetics of REM$_{105\,\mu m}$ ($k_{DET,105\,\mu m}/k_{105\,\mu m}$) was also shown. Notably, TA is relatively resistant to DET reaction but reacts readily with •OH ($k_{\bullet OH,\ TA} = 4.4 \times 10^9$ M$^{-1}$s$^{-1}$)[49]. Interestingly, TA exhibits the lowest enhancement factor of 0.3, which means that the reaction kinetics on REM$_{7\,\mu m}$ was even lower than that on REM$_{105\,\mu m}$. With the improved contribution of DET, the enhancement factor drastically increases. For instance, about 6-fold higher oxidation kinetics was achieved on REM$_{7\,\mu m}$ in comparison to that on REM$_{105\,\mu m}$ for *o*-cresol, with the DET route accounting for 44.1% of the total $k_{105\,\mu m}$. Notably, further increase of DET contribution did not lead to extra improvement on the enhancement factor (e.g., citric acid). Although the contribution of the DET route on REM$_{7\,\mu m}$ is generally higher than on REM$_{105\,\mu m}$, the enhancement factor plotted with respect to $k_{DET,7\,\mu m}/k_{7\,\mu m}$ shows a similar trend (Supplementary Fig. 23). Combined with the above results, it is reasonable to conclude that DET is the main reaction pathway on REM$_{7\,\mu m}$, while •OH is largely responsible for the oxidation reactions on REM$_{105\,\mu m}$.

Thereafter, the •OH generation was quantitatively analyzed by detection of 2-hydroxyterephthalic acid (HTA), a product of the reaction between TA and •OH[50]. HTA concentration reached 1.76 μM on REM$_{105\,\mu m}$, which is much higher than that (0.38 μM) on REM$_{7\,\mu m}$ (Fig. 3d). This result indicates that •OH generation is more favored on REM with larger pores, aligning well with the ESR measurement. The same trend can be observed in the production of 7-hydroxycoumarin (Supplementary Fig. 24), which is another probe for •OH[28]. Although the production of HTA on REM$_{7\,\mu m}$ is only 22% of that on REM$_{105\,\mu m}$, it is important to note that the absolute contribution of radical oxidation to the 4-CP oxidation on REM$_{7\,\mu m}$ is higher than that on REM$_{105\,\mu m}$ (Fig. 3b). This can be attributed to the different mechanisms of the two methods (i.e., quenching experiment and probe test). The HTA production experiment was less affected by mass transfer due to the lower HRT and resistance to DET reaction. However, in the quenching experiment, the impact of DET process on the radical reaction cannot be neglected. Specifically, the reactants have been heavily depleted by the DET reaction in smaller pores, which would improve the absolute contribution of radical oxidation[51,52]. In addition, the •OH production and its contribution to electrooxidation were measured in a recirculation mode[53]. REM$_{105\,\mu m}$ demonstrates a superior capability in TA degradation and HTA production (Supplementary Fig. 25), with an HTA generation rate four times higher than REM$_{7\,\mu m}$. Regarding the degradation of 4-CP, the pseudo-first-order kinetic constant on REM$_{7\,\mu m}$ is three times higher than that on REM$_{105\,\mu m}$, while the contribution of radical oxidation on REM$_{7\,\mu m}$ is much smaller

(Supplementary Fig. 26). Overall, the results obtained under the recirculation mode are well consistent with those of the single-pass mode.

The electrode potentials during the degradation experiments were also recorded and are presented in Fig. 3d. Although the REM with larger pores possessed a higher applied potential, all the electrodes meet the potential requirement for the generation of •OH (ca. 2.8 V$_{RHE}$)[54]. Notably, the production of HTA on REM$_{7\,\mu m}$ is still lower than that on REM$_{105\,\mu m}$ even at the same potential (Supplementary Fig. 27). In this regard, we assume that the lower generation of •OH may be attributed to the nonuniform potential distribution inside the microchannels, which will be discussed in the following section.

### Simulation of •OH generation in microchannels

To understand the reaction mechanisms of REMs, we conducted finite element simulations to study the electrochemical process in membrane pores (Supplementary Fig. 28). Four types of cylindrical microchannels with diameters of 7, 17, 45, and 105 μm were modeled. A schematic illustration of the channel$_{105\,\mu m}$ (the subscript indicates the diameter of the channel) is displayed in Fig. 4a. The inner surface of the channel was set as the conductive electrode, while the counter electrode was perpendicular to the channel. The inlet of the electrolyte flow was at the far end of the channel, which is the same as in the experiments.

Compared to larger channels, there is a substantially larger number of small channels, leading to higher surface area. Therefore, the surface potential and current density of channel$_{7\,\mu m}$ is relatively low (Fig. 4b and Supplementary Fig. 29). It should be noted that the potential and current are unevenly distributed in the channels. The potential value at the far end of the channel is notably lower than that at the near end, accompanied by the lower current. For instance, the surface potential at the far end ($d = 3$ mm) of channel$_{105\,\mu m}$ is 2.86 V, which is 0.27 V lower than that at the near end ($d = 0$ mm). The nonuniformity becomes more pronounced in channels with small diameters, as the potential difference in the two ends of channel$_{7\,\mu m}$ is 0.47 V. In addition, the current density at $d = 0$ mm (0.45 mA cm$^{-2}$) is 15-fold higher than that at $d = 3$ mm (0.03 mA cm$^{-2}$) for channel$_{7\,\mu m}$ (Supplementary Fig. 29). In comparison, a 5-fold current increase (from 0.31 to 1.45 mA cm$^{-2}$) is observed for channel$_{105\,\mu m}$. Such nonuniformity can be attributed to the varied electrical resistance of the solution at different distances, as well described in the transmission line theory[8,55]. The potential and current distributions were qualitatively visualized using Pb$^{2+}$ electrodeposition imprinting[56] (Supplementary Fig. 30). Although PbO$_2$ can be deposited throughout the entire length of REMs, its distribution in REM$_{7\,\mu m}$ is rather nonuniform, indicating the unevenly distributed surface current (Supplementary Fig. 29).

It is widely recognized that a relatively high potential threshold (ca. 2.80 V$_{RHE}$)[21,49] is required to induce the generation of •OH. In this regard, most of the surfaces in channel$_{7\,\mu m}$ are incapable of producing •OH. In contrast, channel$_{105\,\mu m}$ with relatively high potential is more electro-active. To quantitatively analyze the available amount of •OH, simulations were carried out by adding TA to the electrolyte as a radical trapping agent, where the selectivity towards HTA was estimated as 8.8% (Supplementary Fig. 31). The calculated HTA concentration in the effluent are shown in Fig. 4c, where the experimental data in REMs are also shown for comparison. It can be observed that the simulated results are in good accordance with the experiments. For instance, an HTA concentration of 1.65 μM was obtained on the channel$_{105\,\mu m}$ model, which is close to the experimental value (1.76 μM) on REM$_{105\,\mu m}$. The consistency between simulations and experiments validates the high accuracy of the models.

### Simulation of •OH and reactant distribution in microchannels

Based on the above analysis, a composite model was constructed to simulate reactions in the confined microchannels, where mass

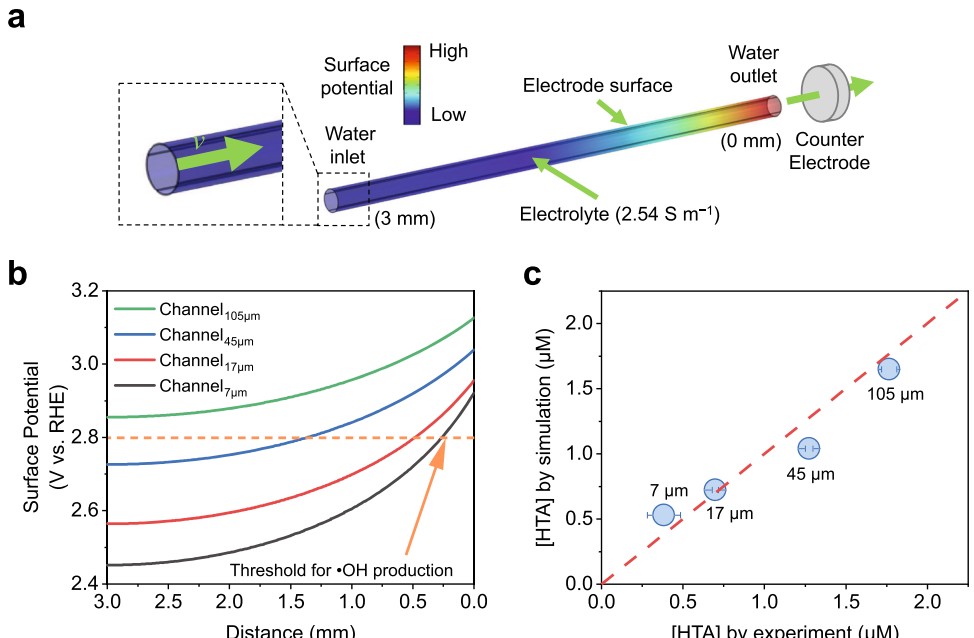

**Fig. 4 | Multiphysics simulation results of the channels. a** Schematic diagram of the model. **b** The surface potential distribution along with the distance to the near end. **c** Correlation between simulated HTA production and experimental HTA production. HRT in the experiment and the simulation: 2.7 s. Error bars represent the data from duplicate tests.

transfer, DET reaction, and radical oxidation were simultaneously considered. To better discuss the DET and •OH oxidation processes, 4-CP was selected as the reactant. Three reactions involving •OH were considered in the simulation: (i) radical oxidation of 4-CP, (ii) radical oxidation of the degradation products of 4-CP[57], and (iii) self-quenching of •OH[58,59]. The concentration distribution of •OH in a cross-section ($d$ = 0.1 mm in Fig. 4a) is shown in Fig. 5a. Due to its high reactivity with organics and short lifetime, •OH radicals are located in the surface region and decrease exponentially with distance from the channel surface. The derived •OH layer thickness (<1 μm) is in agreement with previous studies[58,60]. To quantify the spatial confinement effect in the simulated channels, a reaction layer for radical oxidation was defined as the region where •OH exists at a concentration of >1 nM. Although the lower current density in channel$_{7\,μm}$ leads to a thinner •OH reaction layer (96 nm) compared with that in channel$_{105\,μm}$ (195 nm), the reaction region in channel$_{7\,μm}$ occupies a larger proportion of pore volume (5.2%). In contrast, the •OH reaction region in channel$_{105\,μm}$ only accounts for 0.7% of the volume. The larger reaction region in channel$_{7\,μm}$ suggests that radical oxidation can also benefit from the size reduction, which explains the higher absolute contribution of radical oxidation to the 4-CP oxidation on REM$_{7\,μm}$ (Fig. 3b) despite the lower •OH production (Fig. 3d). However, the production of •OH was almost infeasible at $d$ >0.5 mm for channel$_{7\,μm}$ (Supplementary Fig. 32), thus countering the positive effect of high •OH occupation.

The concentration profile of 4-CP in the same cross-section is shown in Fig. 5b, with 100 μM 4-CP existing in the influent. In a relatively small channel (e.g., channel$_{7\,μm}$), the concentration of 4-CP near the surface is almost identical to the value at the center, whereas an obvious concentration gradient can be observed in channel$_{105\,μm}$ due to the poor mass transfer rate. The thickness of the diffusion layer, derived from the linear region of the concentration gradient, was estimated to be 3 μm in channel$_{7\,μm}$, which is close to the radius of the channel. In comparison, a much thicker diffusion layer of 37 μm was observed in channel$_{105\,μm}$. Nevertheless, the diffusion layer thickness in channel$_{105\,μm}$ is smaller than that of traditional plate electrodes (Supplementary Fig. 33), with the latter being as large as a few hundred

micrometers[61,62]. According to Fick's law, the mass transfer of reactants can be dramatically enhanced by minimizing the diffusion layer thickness, which mitigates the concentration polarization problem of reactant molecules. In this regard, such a spatial confinement effect gives an edge to the oxidation of organics in competition with side reactions (e.g., oxygen evolution reaction) on the electrode surface. As a result, the 4-CP concentration is rapidly depleted to zero along the axial direction of channel$_{7\,μm}$, with almost no concentration gradation between the center and surface of the channel (Supplementary Fig. 34 and Movie 1). In contrast, the residual 4-CP concentration in the effluent of channel$_{105\,μm}$ is calculated to be as high as 45.8 μM, with strong concentration polarization near the surface. This simulation of concentration distribution also explains the result in Supplementary Fig. 15. It shows that as the current increases, the reaction on REM$_{7\,μm}$ is enhanced to a greater extent than that on REM$_{105\,μm}$.

By analyzing the effluent concentration of 4-CP in the microchannel model reactors, we examined the degradation performance under different HRTs (Fig. 5c and Supplementary Fig. 35). The simulated data are in agreement with the experimental results presented in Fig. 3a. The model reactor with smaller pore diameters shows significantly faster 4-CP degradation. For instance, at a current density of 19.7 mA cm$^{-2}$, over 99.8% of 4-CP is eliminated within 13.6 s by channel$_{7\,μm}$, which is much higher than that (66.7% in 13.6 s) by channel$_{105\,μm}$. The fitted kinetic constants in channel$_{7\,μm}$ are 18.3 min$^{-1}$, which is comparable to the experimental value (18.9 min$^{-1}$) of REM$_{7\,μm}$ (Fig. 3b). A strong agreement was shown between the simulation results and the experimental data at different current densities, confirming the effectiveness of our model.

## Discussion

This study establishes a systematic methodology to explore electrochemical reactions in confined microchannels. The significance of the spatial confinement effect in altering the reaction kinetics and even the pathways was demonstrated. By combining the experimental and computational results, we found contrasting trends of DET reaction and radical oxidation affected by the spatial confinement effect. The diffusion layer of reactants was greatly compressed in microchannels,

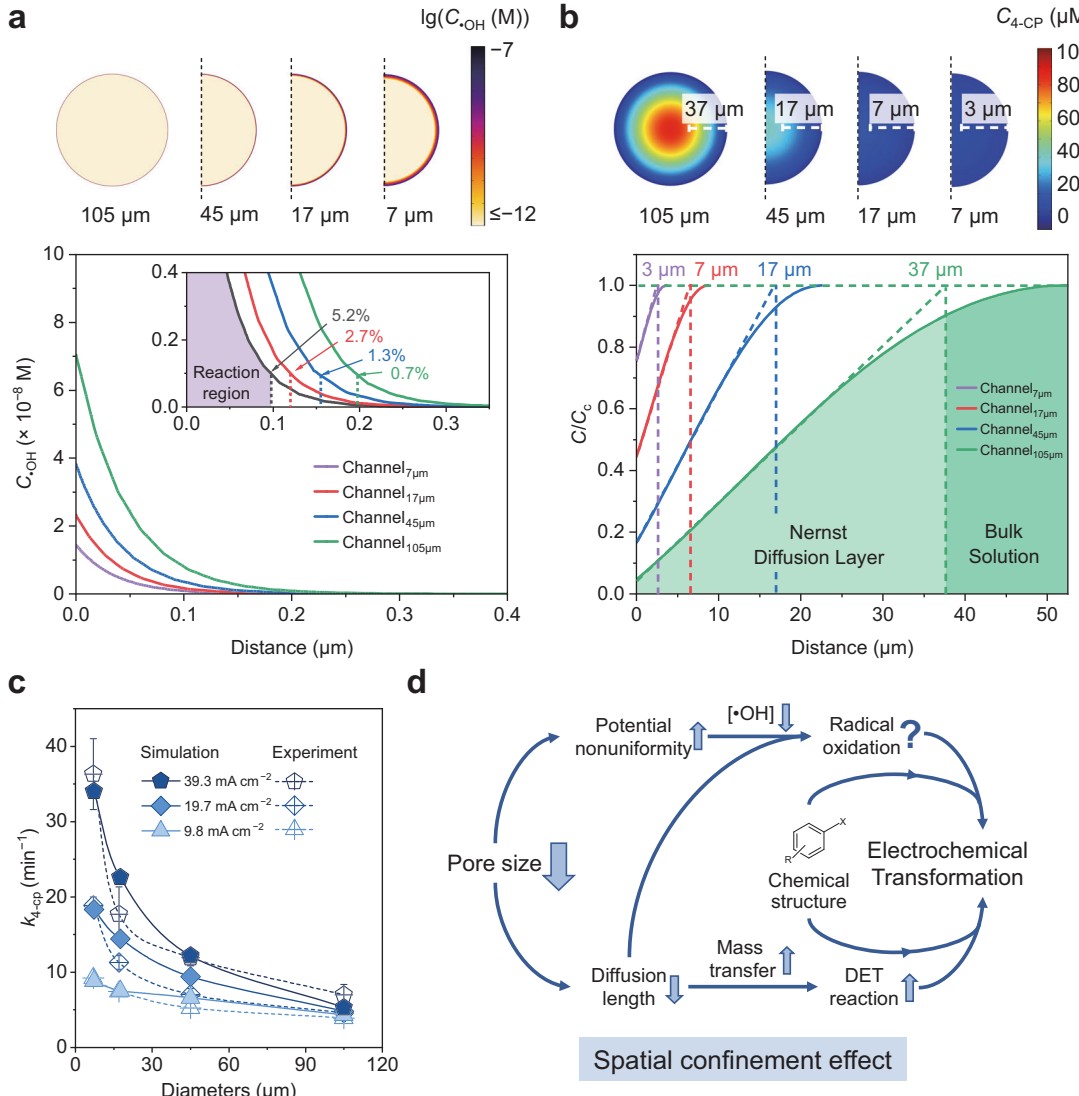

**Fig. 5 | Multiphysics simulation results of the microreactors. a** Visualized •OH distribution in the cross-section (*d* = 0.1 mm) (top) and the corresponding •OH concentration as a function of distance from the channel wall (bottom). The inset image shows the magnified •OH profile and volume ratio of the reaction region. **b** Visualized 4-CP distribution in the cross-section (*d* = 0.1 mm) (top) and the corresponding 4-CP concentration as a function of distance from the channel wall (bottom). $C_c$ represents the concentration of 4-CP at the center of the channel. The HRT in the simulation is 9.1 s. **c** Simulated pseudo-first-order kinetic constants of 4-CP degradation. Error bars of the experimental results represent the data from duplicate tests. **d** Mechanistic illustration of the spatial confinement effect.

leading to accelerated mass transfer and alleviated concentration polarization. As a consequence, a reaction kinetic as high as 18.9 min⁻¹ was achieved for the degradation of organic pollutant (4-CP) on REM$_{7\mu m}$, which is four times higher than that on REM$_{105\mu m}$. Additionally, the nonuniform potential distribution in small channels inhibits the generation of •OH, thereby counteracting the positive effect of enhanced mass transfer.

Based on our experiments and simulations, a mechanistic insight into the spatial confinement effect was obtained (Fig. 5d). Owing to its high reactivity and short lifetime, oxidative •OH is distributed in a thin layer near the channel wall, which is the reaction hotspot for both DET and radical oxidation. By reducing the channel diameter, the diffusion length of reactants can be greatly reduced, leading to accelerated mass transfer and alleviated concentration polarization near the pore surface. Consequently, the interfacial coupling of electrons, reactants, and •OH is strengthened. However, the nonuniformly distributed potential within small channels inhibits the generation of •OH, which may hinder the degradation of refractory contaminants but increase the selectivity towards DET reactions. Additionally, the simulation also

revealed that the electrooxidation performance might not continue to improve if the pore size is further reduced due to the inhibition of •OH generation (Supplementary Fig. 36). Furthermore, increasing the mass transfer efficiency becomes challenging as the concentration polarization has already been significantly alleviated in REM$_{7\mu m}$ (Supplementary Fig. 34). These findings suggest that there is an optimal pore size for REMs, which is consistent with the results of previous studies[44,63]. Moreover, the potential distribution is also influenced by the properties (e.g., transfer coefficient, onset potential for oxygen evolution, and tortuosity) and operational parameters (e.g., applied current) of the electrode, as indicated by the simulations (Supplementary Fig. 37). This suggests that reactions within the pores can be precisely controlled by regulating the electrode structure and operational conditions. In this regard, cost-effective REM systems with high reaction kinetics and selectivity can be achieved by profoundly understanding the spatial confinement effect.

In this work, we report that the electrochemical reaction mechanism inside the confined micropores is significantly affected by the pore size. Although factors such as the irregular pore structure and

generation of oxygen bubbles were not taken into consideration in the simplified model, its accuracy in elucidating spatially confined oxidation processes is adequate. Our findings provide useful guidance for the design of cost-effective REMs and the regulation of reaction pathways. With the rapid development of electrocatalysis, the confinement effect in REMs exhibits promising application potential in environmental remediation and chemical synthesis. The impact of local pH and surface charge of reactants on the electrochemical reactions is a good topic that needs further investigation. Future work is also required to explore the confinement effect in nanosized channels by a better design of the porous scaffold. In addition, in operando visualization of the electrochemical processes in the microchannels is another imperative task to better reveal and regulate the spatial confinement effect. Given that electrocatalytic reduction is an emerging technique for wastewater remediation and resource recovery, applying such a methodology to the cathodic systems is another important area of research. In addition, applying such a methodology to photo(electro)catalytic systems is attractive to gain insights into the direct hole transfer path and •OH radical path.

## Methods

### Materials

All chemicals were used as received, with details provided in Supplementary Information. Porous Ti substrates with a thickness of 3 mm were purchased from Nanjing Shinkai Filter Co., Ltd (China), which were fabricated by compressing Ti particles under high pressure. The pore size of the Ti substrate was controlled by the particle size[14]. All solutions were prepared using deionized water purified by a Milli-Q Plus system (Millipore).

### REMs preparation

The Ti substrates were sequentially washed in acetone, ethanol, and deionized water for 30 min, respectively. Next, the substrate was etched in 10% oxalic acid at 100 °C for 1 h. To fabricate $TiO_2$ nanosheets, the substrate was reacted with 5 M NaOH solution at 180 °C for 3 h in a Teflon-lined stainless steel autoclave. Then, the obtained $Na_2Ti_3O_7$-coated Ti substrate was rinsed in 1 M hydrochloric acid for 1 h to generate the $H_2Ti_3O_7$ layer[64]. Finally, the REM was prepared by calcinating Ti substrate at 400 °C for 2 h in air and subsequently annealing it at 500 °C for 6 h in an argon atmosphere. For comparison, REMs with pore diameters of 7, 17, 45, and 105 μm were prepared using the above-mentioned method.

### Characterization

The morphology of the prepared electrode was examined using a scanning electron microscope (SEM, Hitachi SU8000). XRD patterns of the electrode were recorded using a D8 advance X-ray diffractometer (Bruker) with Cu Kα radiation. XPS spectra were acquired on a Thermo ESCALAB250Xi spectrometer with a monochromated Al Kα X-ray source. ESR analysis was conducted using a Bruker A300 spectrometer. The pore size distribution was characterized using the mercury intrusion method (AutoPore IV 9500). The porosity of REMs was determined using Archimedes' drainage method (Supplementary Table 3). All electrooxidation experiments were performed on a CHI660E electrochemical workstation (CH Instruments). All the electrochemical measurements were conducted on a Gamry Interface 1000 electrochemical workstation.

### Mass transfer coefficient determination

The mass transfer coefficient ($k_m$) to the electrode was calculated using a well-established approach[65]. A solution containing 0.1 mM $Fe(CN)_6^{4-}$, 0.2 mM $Fe(CN)_6^{3-}$, and 0.33 M $NaClO_4$ was used as the electrolyte. A dead-end filtration assembly was employed to conduct all the electrochemical experiments, where the membrane flux was controlled using a peristaltic pump (Supplementary Fig. 1b). The

counter and reference electrodes were a Ti wire and Ag/AgCl electrode, respectively. All the electrochemical experiments were performed on a CHI660E electrochemical workstation (CH Instruments) at $30 \pm 1\,°C$. The oxidation current of $Fe(CN)_6^{4-}$ was obtained at 1.91 $V_{RHE}$, with details described in Supplementary Fig. 12.

The observed mass transfer rate ($K_{obs}$) was calculated using Eq. (1):[39]

$$K_{obs} = \frac{I}{zFAC_b} \tag{1}$$

where $I$ is the oxidation current of $Fe(CN)_6^{4-}$ (A), $z$ represents the number of electrons transferred (1 for the oxidation of $Fe(CN)_6^{4-}$), $F$ denotes the Faraday constant (96,500 C mol$^{-1}$), $A$ is the geometry surface area of the electrode ($2.54 \times 10^{-4}$ m$^2$), and $C_b$ is the bulk concentration of $Fe(CN)_6^{4-}$ (0.1 mol m$^{-3}$).

The volumetrically averaged mass transfer coefficient ($k_m A_e$) was calculated using Eq. (2):[42]

$$k_m A_e = \frac{I}{zFV_e C_b} = \frac{K_{obs}}{L} \tag{2}$$

where $k_m$ is the mass transfer coefficient (m s$^{-1}$), $A_e$ denotes the active electrode area per unit volume (m$^{-1}$), $L$ represents the thickness of the porous electrode (m), and $V_e$ is the total volume of the electrode within the reactor ($7.63 \times 10^{-7}$ m$^3$).

### Organics oxidation on REMs

The oxidation experiments of organics were conducted on the reactor similar to $Fe(CN)_6^{4-}$ oxidation, where a $RuO_2$/Ti mesh was used as the cathode. The reaction area, 2.54 cm$^2$ (i.e., 0.9 cm × 0.9 cm × π = 2.54 cm$^2$) was defined by a silicone gasket (thickness: 1 mm). The thickness of REMs was 3 mm. The degradation experiments were performed under a fixed current (50 mA) and single-pass operation mode at $30 \pm 1\,°C$, where the HRT was calculated based on the flux and volume of REMs. The stock solution containing 100 μM reactant (e.g., 4-CP) and 0.33 M $NaClO_4$ was used as the electrolyte (pH = 7). Before each sampling, a fixed membrane flux was maintained until 20 mL effluent was collected to ensure a homogeneous effluent. The organic concentrations were determined using HPLC (Agilent 1260) as described in Supplementary Information.

Pseudo-first-order kinetics was used to fit the degradation of organics according to the following equation:

$$C_t = C_0 e^{-kt} \tag{3}$$

where $t$ is the HRT (s), $C_t$ represents the effluent concentration at a given HRT, $C_0$ is the initial concentration (100 μM), $k$ represents the pseudo-first-order kinetic constant (s$^{-1}$).

### Finite element simulation

The concentration distribution of •OH and reactant in the microchannels was simulated using COMSOL Multiphysics 5.3a. The three-dimensional geometry "CFD (computational fluid dynamics)" module was used to solve the flow velocity and streamline distribution in the microchannels. The simulation of potential and current distribution was conducted based on a modified transmission line model[8,55], where the "secondary current distribution" module was used. Four crucial reaction processes including DET oxidation of the target organic, radical oxidation of the target organic, radical oxidation of intermediate products, and self-quenching of •OH were constructed using the "chemistry" module to simulate reactions in confined microchannels. The •OH production rate was proportional to the anodic current density, as estimated in the TA degradation experiment

(Supplementary Fig. 38). The "transport of diluted species" module was used to solve the concentration distribution. Microchannels with lengths of 3 mm and diameters of 7, 17, 45, and 105 μm were constructed. The detailed parameters are presented in Supplementary Information.

## Data availability

The data supporting the findings of this work are available within the article and its Supplementary Information files. Any additional data reported in this work are available from the authors on request.

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

## Acknowledgements

This work was financially supported by the National Natural Science Foundation of China (No. 52125003 received by C.H., 52100109 received by Z.G.), Chinese Academy of Sciences (No. 032GJHZ2023001MI received by J.Q.), and the Yangtze River Joint Research Phase II Program (No. 2022-LHYJ-02-0303 received by C.H.).

## Author contributions

Z.G. and J.Q. designed the experiments; Y.K. and Z.G. performed the research; Y.K., Z.G., B.M., and W.Z. analyzed the data; Y.K., Z.G., C.H., and J.Q. wrote the paper; J.S., X.H., and W.C. discussed the results. All authors discussed, commented on, and revised the manuscript.

## Competing interests

The authors declare no competing interest.
