## [Peer Review File · Nature Communications]

Unveiling the spatially confined oxidation processes in reactive electrochemical membranesREVIEWER COMMENTS

Reviewer #1 (Remarks to the Author):

The manuscript NCOMMS-23-19868 developed a model to explore the mechanism of direct electron transfer and radical oxidation in spatially confined reactive electrochemical membranes (REM). The authors revealed that the ununiform distribution of voltage in the REM highly affected the reaction mechanism for the removal of organic contaminants. This observation can benefit the future design of the REM system to enhance its performance in water treatment. Overall, this work meets the standard to be published for Nature Communication and will increase the readership. However, some sections of this manuscript require further clarification or modification to increase the quality. Detailed comments are listed below.

- 1) Line 89: The pores in REM are in a wide size distribution in Figure 1. What is the percentage threshold to determine the dominant size denoted for each material? For example, REM 7 μ m featured a large portion of other sizes of pores.
- 2) Lines 85-107: The authors have conducted extensive characterization of REMs using different techniques. However, these results are not properly discussed. More detailed discussion or explanation is needed to increase the quality and prevent any confusion or misunderstandings. This comment is for SI Fig 2 to SI Fig 10. If adding discussion will be over the word limit, the authors can add them in the SI.
- 3) Line 101: How the 4 times higher surface area was determined? In SI Fig. 9 or SI Table 1? There are different data for REM area. What term was used to compare? Again, more detailed explanation is needed.
- 4) Line 104: How do the authors make this conclusion or statement? Why surface area is related to the current difference?
- 5) Line 106: A duration of 15 hours is short to test the stability of a REM. Also, the electrolyte condition is mild for the stability test. It is suggested to conduct an accelerated service life test using a current density of 0.5 A/cm² in 0.5 M H₂SO₄ solution at 30 °C. The method can be found here <https://doi.org/10.1016/j.electacta.2020.135634>.
- 6) Fig. 2b and SI Fig. 13: These two figures demonstrate different mechanisms. Based on Fig. 2b, the effect of surface area increase is irrelevant to water flux, because the green area starts from flux of zero. However, in SI Fig. 13, when water flux is less than 0.1 cm/s, e.g., the ratio is less than 4, the enhancement is because of surface area increase, while when the flux is higher than 0.1 cm/s, both diffusion enhancement and surface area increase contribute. The reviewer believes it may not be valid to differentiate the roles of surface area and diffusion. To some extent, these two factors are correlated.
- 7) Line 125: One assumption here that the authors did not mention is that the surface area is linearly correlated to the current increase. Is this assumption valid? Any reference for this assumption? Hence, it is unsure in Figure 2b if the 4 times increase in surface area can cause a 4 times increase in current density.
- 8) Lines 166-171: In the text of Energy Consumption Calculation, the current is 50 mA. However, in Fig. 3, the current density is 19.7 mA/cm² (i.e., 1.25 cm * 1.25 cm * 19.7 mA/cm² = 97 mA). These two values are different. Which number was used to calculate energy consumption?
- 9) SI Line 230: what is the role of the dash line in the figure?
- 10) It is suggested to add the unit for parameters in the description for all equations in the main text and the SI to increase readability.
- 11) SI Fig. 23: How was the percent yield obtained?
- 12) Fig. 5d: it is suggested to remove the Taijitu sign.
- 13) Line 363: REM characterization should be mentioned in the main text. It is totally fine to keep the details in the SI. However, none is stated in the method section of the main text.
- 14) Line 377: The authors are suggested to provide details about how the REMs with different pore diameters were fabricated. How was the pore size controlled?
- 15) Line 390: It should be "(1)".
- 16) Line 418: DET reaction needs to be clarified. What DET reactions have been considered?

Reviewer #2 (Remarks to the Author):

The authors reported an interesting study on the electron transfer and radical oxidation processes in the microchannels of REM, which are important topics in electrocatalysis and environmental remediation. Both experiments and simulations were conducted to investigate the reaction processes. The authors demonstrate that not only the oxidation kinetics but also the electron transfer mechanism can be largely impacted by the pore diameter. The methodology and results of this study are enlightening and could attract wide attention. The data provided are comprehensive and convincing. I suggest the manuscript be accepted for publication after carefully considering the following points.

1. Defective TiO₂ is employed as the electrocatalyst of the REM electrodes. Meanwhile, TiO₂ is an extensively studied photocatalyst and could generate hydroxyl radicals under light irradiation. The impact of light on the electrochemical oxidation processes needs to be clarified.
2. The authors employed 200 mM TBA to quench •OH radicals and determine the contribution of different electron transfer routes. It's suggested to provide the reaction kinetics at different concentrations of TBA to prove that 200 mM TBA is high enough to fully quench the radicals.
3. The generation of •OH radicals is detected by using TA and coumarin as the probing molecules, respectively. Although these two methods are approved, the DMPO trapping method is more acceptable. I suggest the authors provide the ESR signals of DMPO-OH on different REMs.
4. Besides the removal of specific pollutant molecules, the complete oxidation of organics to CO₂ is also critical for water purification. The representative total organic carbon (TOC) values and the corresponding current efficiencies for organic oxidation are suggested to be provided.
5. Both the experiments and simulations demonstrate that the oxidation of 4-CP can be drastically enhanced by reducing the pore sizes of REM. However, the lowest pore diameter investigated in this study is 7 μm, which is not low enough. Although further reducing the pore diameter may be not easy in the experiment, I suppose it is feasible by simulation. This work would be more attractive if the authors could "predict" what would happen when the pore diameter is further reduced.
6. Some of the important parameters for the simulation of HTA generation (Fig. 4c) are not shown, such as the diffusion coefficient of TA, the reaction kinetic constant of TA and OH radical.
7. As the current methodology is sophisticated, I think its application to the cathodic systems should be feasible. A brief discussion on such a possibility can be helpful.
8. There are some format errors in this manuscript. For instance, the sequence number of the equation in line 390 should be 1; the unit of the x-axis in Fig. S12a seems to be wrong. The authors should carefully check the whole manuscript to avoid such errors.

Reviewer #3 (Remarks to the Author):

The manuscript entitled "Unveiling the spatially confined oxidation processes in reactive electrochemical membranes" provides interesting data and the authors found that both the reaction kinetics and the electron transfer pathway can be effectively regulated by the spatial confinement effect. The reviewers believe that the manuscript has positive implications for the field of REM wastewater treatment. Here are my comments:

1. Microfluidic has some unique fluid properties, such as laminar flow and droplets. I do not really agree that REM can be considered as a microfluidic platform.
2. Change mL cm⁻² s⁻¹ to m s⁻¹
3. Fig. S7, the XPS O1s deconvolution is not reasonable and need to be re-fitted.

4. I cannot agree the statement that the REM electrode has high electrochemical stability. As shown in Supplementary Figure 10, the anode potential has increased over 1 V during a 15 h testing at a constant current of 20 mA cm⁻², suggesting that anode oxygen evolution activity has decayed significantly. The reviewer recommends that all REM electrodes should be treated with continuous electrolysis until their activity is stabilized prior to use.

5. Fig. 2b and Supplementary Figure 13, since the Fe(CN)₆⁴⁻ electron transfer reaction is carried out under mass transfer control, there is no comparability between the reaction surface area and the reaction current. We cannot say briefly which is the enhanced reaction activity due to the enhancement of reaction area.

6. Supplementary Figure 12, how did the authors obtain the limiting currents? Being in the reaction control region, the reaction limiting current should be linearly related to the number of active sites, i.e., the electroactive area of the REM electrode. The authors need to reevaluate the calculation methods they use.

7. Fig. 2, the achieved k_m values (Y-axis, which should actually be k_{obs}) should be compared with the literature.

8. Figure 3b clearly shows that the absolute contribution of hydroxyl radicals to 4-CP oxidation on REM7 μ m exceeds the total amount of 4-CP oxidized by REM105 μ m (including DET oxidation and hydroxyl radicals mediated oxidation) by a factor of 3, indicating that the hydroxyl radical yield of REM7 μ m is much higher than that of REM105 μ m. However, this contradicts many of the data in the manuscript, as shown in Fig. 3c,3d, Supplementary Figures 18-20, etc.

9. Many experiments, for which the authors only give data under 20 mA cm⁻². The reviewer suggests that the authors perform more experiments at lower as well as higher current densities (electrode potentials).

10. Frankly, the reviewer has doubts about the simulation results of 4-CP degradation kinetics (2.24 s⁻¹ vs 2.25 s⁻¹) and the TA reaction to produce HTA (1.79 μ M vs 1.76 μ M, Fig. 4b). As we know, REM substrate (porous titanium filter) used in this study is made of sintered titanium particles, which determines that the pore structure inside the REM electrode is very complex and variable, and the pore size distribution is also very heterogeneous (Fig. 1b). Although the assumption of microtubules can simplify the computational model to the maximum extent (There is a huge difference between the model and the actual situation), it is often difficult to give accurate quantitative results. In addition, the bubbles generated by the electrolysis process can also affect the simulation results by perturbing the boundary layer as well as the distribution of hydroxyl radicals. Supplementary Figure 23 shows that the selectivity towards HTA produced from the oxidation of TA under REM with different pore size is also variable (although the authors chose a mean value of 7% for the simulation). What the authors need to explain is how they achieved simulation results (Fig. 4b) that are so close to the accurate experimental results using a distortion model with many simplifications.

Point-by-point response to the reviewers' comments

Title: “*Unveiling the spatially confined oxidation processes in reactive electrochemical membranes*”

Manuscript ID: NCOMMS-23-19868A

We sincerely thank all reviewers for their valuable comments and suggestions, which are certainly helpful in improving the quality of our work. We have carefully and systematically responded to all the points raised. The reviewers' comments are in bold italic font and our revisions are in blue font. We have also highlighted the revised text in blue in the main text. Provided below are our detailed responses to each point.

REVIEWER #1 (Comments to authors)

General Comment: The manuscript NCOMMS-23-19868 developed a model to explore the mechanism of direct electron transfer and radical oxidation in spatially confined reactive electrochemical membranes (REM). The authors revealed that the ununiform distribution of voltage in the REM highly affected the reaction mechanism for the removal of organic contaminants. This observation can benefit the future design of the REM system to enhance its performance in water treatment. Overall, this work meets the standard to be published for Nature Communication and will increase the readership. However, some sections of this manuscript require further clarification or modification to increase the quality. Detailed comments are listed below.

Thank you for your recognition of this work and your constructive suggestions on this manuscript. Your insightful comments will greatly help us improve the quality of this paper. A point-by-point response to your comments is as follows.

Comment 1: Line 89: The pores in REM are in a wide size distribution in Figure 1. What is the percentage threshold to determine the dominant size denoted for each material? For example, REM 7 μ m featured a large portion of other sizes of pores.

We appreciate your valuable comment. The determination of pore size is critical

for the theoretical simulations in this work. Although fabricating stable electrodes with uniform channels is highly desired, it is technically challenging. In this study, we used porous titanium fabricated from titanium particles as the electrode substrate, which allowed for adjustable pore sizes to the highest degree while ensuring electrode stability. However, due to the limitations of the electrode substrate material, the uneven distribution of pores in the electrode is unavoidable.

Here, the pore size denoted for each REM was determined by the size that accounted for the largest percentage in the differential mercury intrusion of REMs (Fig. 1b). Specifically, the predominant pore sizes for REM_{7 μ m}, REM_{17 μ m}, and REM_{45 μ m} were 7, 17, and 45 μ m, respectively. For the electrode with the largest pore size (REM_{105 μ m}), a high proportion of pore size was observed in the range of 90 – 120 μ m. Therefore, this electrode was denoted as REM_{105 μ m}, which is the average of 90 and 120 μ m.

To quantitatively describe the proportion of the predominant pore size, we conducted calculations to determine the volume percentage of the predominant pore size within the range of $\pm 40\%$, using the data obtained from mercury intrusion measurements. As depicted in the revised Supplementary Fig. 4, the volume percentages of the predominant pore size were found to be in the range of 45% to 48%, indicating that the pore sizes we have identified are indeed representative.

We have added the relevant statements in the revised main text.

Line 86 – 90 in the revised manuscript:

“The REMs were denoted by their predominant pore sizes, as REM_{7 μ m}, REM_{17 μ m}, REM_{45 μ m}, and REM_{105 μ m}, respectively (Fig. 1b). Over 45% of the total pore volume falls within the $\pm 40\%$ range around the predominant size, indicating that the selected pore size is representative (Supplementary Fig. 4).”

Page 9 in the revised Supplementary Information:

“

Supplementary Fig. 4. The pore size distribution of REMs by mercury intrusion method. The predominant pore sizes in the REMs were 7, 17, 45, and 90 – 120 μm , respectively. These electrodes are denoted as REM_{7 μm} , REM_{17 μm} , REM_{45 μm} , and REM_{105 μm} , respectively. Note that REM_{105 μm} was denoted using the arithmetic mean of the two predominant pore sizes. Over 45% of the total pore volume falls within the $\pm 40\%$ range around the predominant size, indicating that the selected pore sizes are representative.”

Comment 2: Lines 85-107: *The authors have conducted extensive characterization of REMs using different techniques. However, these results are not properly discussed. More detailed discussion or explanation is needed to increase the quality and prevent any confusion or misunderstandings. This comment is for SI Fig 2 to SI Fig 10. If adding discussion will be over the word limit, the authors can add them in the SI.*

We appreciate your important comments. The structural characterization is indeed essential to improve the comprehensiveness of the manuscript and prevent any possible confusion. We have provided a detailed discussion of this part in our revised manuscript, which we believe will effectively prevent any confusion that may have arisen in the previous version. We have added the relevant statements in the revised manuscript and supporting information.

Line 83 – 108 in the revised manuscript:

“To quantitatively assess the spatial confinement effect, we constructed a series of REMs using Ti membranes as the conductive substrates (Supplementary Fig. 2). The interconnected pore network served as water channels, on which defective TiO₂ nanosheets were fabricated as electrocatalysts (Supplementary Fig. 3). The REMs were denoted by their predominant pore sizes, as REM_{7μm}, REM_{17μm}, REM_{45μm}, and REM_{105μm}, respectively (Fig. 1b). Over 45% of the total pore volume falls within the ±40% range around the predominant size, indicating that the selected pore size is representative (Supplementary Fig. 4). It should be noted that the Brunauer–Emmett–Teller (BET) surface area of REMs increases with the reduction of pore size (Supplementary Table 1). For instance, the surface area of REM_{7μm} was determined to be 10.85 m² g⁻¹, which is four times that of REM_{105μm}. Such a trend in BET surface area is consistent well with the electrochemical double-layer capacitance (Supplementary Fig. 5) and mercury intrusion results (Supplementary Table 1).

Meanwhile, pore size showed little difference in the crystalline structure of REMs, as revealed using X-ray diffraction (XRD, Supplementary Fig. 6). Following annealing in an argon atmosphere, the color of REMs changed to a grayish black, indicating the existence of crystal defects (Supplementary Fig. 7). The electronic structure of REMs was characterized using electron spin resonance (ESR) spectroscopy and X-ray photoelectron spectroscopy (XPS). A typical sign of oxygen vacancies³² at $g = 2.003$ was observed (Supplementary Fig. 8), indicating the presence of defects in the TiO₂ nanosheets. After annealing in an argon atmosphere, the oxygen vacancy increased, as shown in the O 1s XPS band (Supplementary Fig. 9a and b)³³. Additionally, the Ti 2p peak and the valence band edge shifted toward lower binding energy (Supplementary Fig. 9c and d), indicating an increased electron density with the introduction of oxygen vacancy, which substantially improves the conductivity and electrochemical activity^{34,35}.”

Line 123 – 126 in the revised manuscript:

“Moreover, steady potentials and electrooxidation performance were maintained throughout the experimental duration at a fixed current density, underscoring the capability of REMs in serving as model electrodes (Supplementary Fig. 11).”

Page 14 in the revised Supplementary Information:

“

Supplementary Fig. 9. XPS spectra of REM_{7µm} before and after annealing in an argon atmosphere. O 1s XPS spectra of REM_{7µm} before (a) and after (b) annealing in an argon atmosphere. (c) Ti 2p XPS spectra. (d) valence band XPS spectra. For the O 1s spectra, the peaks at 530.3, 531.5, and 532.6 eV were attributed to lattice oxygen (O_L), adsorbed oxygen (O_{ads}), and surface oxygen (H₂O), respectively^{16,17}. The increase of the O_{ads} demonstrates the existence of oxygen vacancies accompanied by localized electrons richness¹⁸. The Ti 2p peak shifted to a lower binding energy by -0.15 eV, indicating the lattice Ti⁴⁺ atoms were partly reduced to Ti³⁺. Consistent with the deconvolution of the O 1s XPS band, the unsaturated Ti³⁺ further suggested the existence of oxygen vacancies¹⁹. The position of the valence band edge shifted from 2.70 eV to 2.50 eV, showing a narrowed band gap after the thermal treatment²⁰.”

Page 16 in the revised Supplementary Information:

“

Supplementary Fig. 11. Potential-time curve and 4-CP degradation performance of REM_{7µm} at a current density of 19.7 mA cm⁻². Electrolyte: 0.33 M NaClO₄. All prepared electrodes were subjected to pre-electrolysis for 3 hours. Subsequently, all experiments on each electrode were performed within 180 minutes after the pre-electrolysis.”

Comment 3: Line 101: How the 4 times higher surface area was determined? In SI Fig. 9 or SI Table 1? There are different data for REM area. What term was used to compare? Again, more detailed explanation is needed.

Thanks for this instructive comment. The specific surface area is an important parameter that affects electrode performance. We used nitrogen adsorption-desorption, mercury intrusion method, and electrochemical double-layer capacitance to compare the surface area of REMs. Specifically, surface areas can be directly derived from the former two methods. Meanwhile, the electrochemical double-layer capacitance is proportional to the surface area, which was calculated based on the results in SI Fig. 9. As shown in SI Table 1, the data from the three methods showed very good consistency.

Ultimately, the Brunauer–Emmett–Teller (BET) surface area derived from nitrogen adsorption-desorption was used for subsequent analysis, and it is well-recognized by the researchers. The results were thoroughly discussed in the revised main text to avoid any possible confusion.

Line 90 – 95 in the revised manuscript:

“It should be noted that the Brunauer–Emmett–Teller (BET) surface area of REMs increases with the reduction of pore size (Supplementary Table 1). For instance, the

surface area of REM_{7μm} was determined to be 10.85 m² g⁻¹, which is four times that of REM_{105μm}. Such a trend in BET surface area is consistent well with the electrochemical double-layer capacitance (Supplementary Fig. 5) and mercury intrusion results (Supplementary Table 1).”

Page 10 in the revised Supplementary Information:

“

Supplementary Fig. 5. The electrical double-layer capacitance (C_{dl}) and corresponding cyclic voltammetry (CV) curves of different REMs. (a) C_{dl} by plotting current variation against the scan rate to fit a linear regression. Cyclic voltammetry (CV) curves of (b)

REM_{7μm}, (c) REM_{17μm}, (d) REM_{45μm}, and (e) REM_{105μm}. C_{dl} values were measured at the potential of 0.89 V vs. RHE. Electrolyte: 0.33 M NaClO₄. Note that C_{dl} value is proportional to the electrochemically active surface area. This result is well consistent with the Brunauer–Emmett–Teller (BET) surface area (Supplementary Table 1), which was used for the analysis in this study.”

Page 40 in the revised Supplementary Information:

“

Supplementary Table 1. The surface area of REMs and corresponding microchannels.

	BET Area ^a	Mercury Intrusion	Cdl ^b
	[m ² g ⁻¹]	Area ^a [m ² g ⁻¹]	[F cm ⁻²]
REM _{7μm}	10.85	9.98	0.41
REM _{17μm}	6.73	6.59	0.26
REM _{45μm}	5.36	5.88	0.21
REM _{105μm}	2.68	2.47	0.10

^aThe Brunauer–Emmett–Teller (BET) surface area and mercury intrusion area reveal the microstructure surface area of REMs and these results show good consistency. The well-recognized BET surface area was used for the subsequent analysis in this study.

^bThe C_{dl} value is proportional to the electrochemically active surface area. This result corresponds well with the BET surface area.”

Comment 4: Line 104: How do the authors make this conclusion or statement? Why surface area is related to the current difference?

We thank the reviewer for this valuable comment. According to the current-potential characteristic (*Electrochemical Methods: Fundamentals and Applications* (Wiley, 2001)), one of the foundational formulas in electrochemistry, the current is related to the surface potential and electrode surface area.

$$i = F A k^0 [C_O e^{-\alpha f(E-E^0)} - C_R e^{(1-\alpha)f(E-E^0)}]$$

F is the faraday constant (96485 C mol^{-1}), A is the surface area of electrode (m^2), C_O is the surface concentration of the oxidative species (mol m^{-3}), C_R is the surface concentration of the reductive species (mol m^{-3}), k^0 is the standard rate constant (m s^{-1}), α is the transfer coefficient, $f = \frac{F}{RT}$, E^0 is the formal potential (V), and E is the electrode potential (V).

The onset potential on REMs is very similar, indicating the overpotential ($E - E^0$) on all REMs are comparable, and cannot account for the large difference of current density on REMs. Meanwhile, there were great differences in the specific surface areas of the REMs. Since REM_{7 μm} has the largest specific surface area, it makes sense that it has the largest current. However, in our system, the current was not directly proportional to the specific surface area. This may be related to the ununiform potential distribution within the pores, which is discussed in detail in Fig. 3.

We have updated the relevant statements and references in the revised main text.

Line 115 – 120 in the revised manuscript:

“The current increased as the pore size decreased, following a similar trend observed for the specific surface area of REMs. However, according to the Butler–Volmer equation³⁷, the current is expected to be proportional to the electrode area, which is not completely consistent with the observed result. This discrepancy may be attributed to the nonuniform potential distribution within the pores, which is discussed in detail below.”

Comment 5: Line 106: A duration of 15 hours is short to test the stability of a REM. Also, the electrolyte condition is mild for the stability test. It is suggested to conduct an accelerated service life test using a current density of 0.5 A/cm² in 0.5 M H₂SO₄ solution at 30 °C. The method can be found here <https://doi.org/10.1016/j.electacta.2020.135634>.

We appreciate your important comments. In general, an accelerated service life test is necessary to verify the stability of electrodes. However, in this study, the REMs we fabricated were only used to investigate the spatial confinement effect in electrooxidation systems. To ensure even coverage of electrocatalyst on the electrode surface and prevent blockage of the pores (*Environ. Int.* **2020**, 140, 105813), we did not cover the electrode surface with the commonly used metal oxide coating layer (e.g., SnO₂-Sb and PbO₂). Actually, the REMs we fabricated in this way do not meet the

requirements for long-term commercial use, but it is sufficient for the investigation of the reaction mechanism.

In this study, all the prepared electrodes need to be treated with pre-electrolysis for 3 hours to reach a stable state. Then we performed three successive 4-CP degradation experiments within a time period of 3 hours. The results demonstrated that the electrodes maintained good stability during the experiment. In this regard, the prepared electrodes were adequately stable for this study.

Thanks again for your valuable suggestions. In future studies, more rigorous tests will be carried out to assess the stability and service time of the REM electrodes.

To clearly demonstrate the satisfactory stability of the REMs in our study, we made modifications to Fig. 11 in the Supplementary Information (previously presented as SI Fig. 10) and updated the corresponding statements in the main text.

Line 123 – 126 in the revised manuscript:

“Moreover, steady potentials and electrooxidation performance were maintained throughout the experimental duration at a fixed current density, underscoring the capability of REMs in serving as model electrodes (Supplementary Fig. 11).”

Page 17 in the revised Supplementary Information:

“

Supplementary Fig. 11. Potential-time curve and 4-CP degradation performance of REM_{7µm} at a current density of 19.7 mA cm⁻². Electrolyte: 0.33 M NaClO₄. All prepared

electrodes were subjected to pre-electrolysis for 3 hours. Subsequently, all experiments on each electrode were performed within 180 minutes after the pre-electrolysis.”

Comment 6: Fig. 2b and SI Fig. 13: These two figures demonstrate different mechanisms. Based on Fig. 2b, the effect of surface area increase is irrelevant to water flux, because the green area starts from flux of zero. However, in SI Fig. 13, when water flux is less than 0.1 cm/s, e.g., the ratio is less than 4, the enhancement is because of surface area increase, while when the flux is higher than 0.1 cm/s, both diffusion enhancement and surface area increase contribute. The reviewer believes it may not be valid to differentiate the roles of surface area and diffusion. To some extent, these two factors are correlated.

We appreciate your important comments and apologize for the ambiguous expressions. In fact, in most cases, both surface area and diffusion contribute to the reaction rate on REMs, as demonstrated by the current-potential characteristic (*Electrochemical Methods: Fundamentals and Applications* (Wiley, 2001)).

$$i = F A k^0 [C_O e^{-\alpha f(E-E^0)} - C_R e^{(1-\alpha)f(E-E^0)}]$$

The surface concentrations (C_O and C_R) are strongly influenced by the mass transfer process. Therefore, it is rather challenging to quantitatively distinguish the contributions of surface area and mass transfer enhancement to the reaction kinetics in our system.

As mentioned earlier, REM_{7 μ m} has four times the area of REM_{105 μ m}, meaning the area can account for at most a 4-fold increase in the reaction rate. The additional enhancement stems from the accelerated diffusion within the pores. In addition, due to reactant depletion and the nonuniformly distributed surface potential, the increase in surface area does not always lead to a proportional current increase, as shown in Fig. 1c. And this will be detailly discussed in Comment 7.

To better illustrate the contributions of mass transfer and surface area, we have decided to remove Supplementary Fig. 13, as it was found to be confusing. We have replotted Fig. 2b with the observed mass transfer rate (K_{obs}) on the y-axis. The value can be calculated from equation (1) in the revised manuscript, which shows the relationship between current, mass transfer, and surface area. The revised figure clearly shows that while surface area can enhance the oxidation reaction to some extent, the

confinement-enhanced diffusion also plays a significant role. Accordingly, the figures and discussions have been revised as follows:

Line 150 – 182 in the revised manuscript:

“To quantitatively investigate the mass transfer process, the observed mass transfer rate (K_{obs}) was determined from the anodic current according to equation (1)³⁹(Fig. 2b). The mass transfer rate can also be expressed as the product of mass transfer coefficient (k_m) and surface area (A_e), which is widely employed and shown in equation (2)⁴². However, in the REM system, the increase in surface area does not result in a proportional improvement in kinetics, as validated by the current curves in Fig. 1c. Two factors can contribute to this. First, the potential distribution within the microchannels is nonuniform, influencing the effective reaction area, and it is discussed in detail below. Second, there is a strong interaction between mass transfer and effective surface area, where one factor substantially affects the other. For instance, the influent $\text{Fe}(\text{CN})_6^{4-}$ ions in $\text{REM}_{7\mu\text{m}}$ can be completely oxidized at a relatively low flux, whereas in the $\text{REM}_{105\mu\text{m}}$ at the identical flux, they cannot be fully oxidized (Fig. 2c). Therefore, the effective surface area of $\text{REM}_{7\mu\text{m}}$ decreases.

Although quantifying the contribution of surface area and mass transfer is challenging, we can roughly categorize the kinetics enhancement into two parts: (i) the maximum A_e contribution (i.e., 4-fold) and (ii) the minimum k_m contribution (Fig. 2b). At fluxes $> 0.5 \times 10^{-2} \text{ m s}^{-1}$, $\text{REM}_{7\mu\text{m}}$ exhibited approximately 7-fold higher K_{obs} than $\text{REM}_{105\mu\text{m}}$. Considering that the surface area of $\text{REM}_{7\mu\text{m}}$ is only 4-fold greater than that of $\text{REM}_{105\mu\text{m}}$ (Supplementary Table 1), the enhanced K_{obs} value of $\text{REM}_{7\mu\text{m}}$ cannot be merely attributed to the relatively large surface area. The accelerated mass diffusion contributed by smaller pores is another important reason. Supplementary Fig. 14 depicts the relationship between $k_m A_e$ and flux for various REMs, where the electrode performance from literature was also presented for comparison. $\text{REM}_{7\mu\text{m}}$ is among the best-performing electrodes with high mass transfer rate and reaction kinetics. This enables rapid contaminant removal and chemical transformation.

Fig. 2 Mass transfer in REMs. a, b Oxidation current of $\text{Fe}(\text{CN})_6^{4-}$ (a) on REMs at different fluxes and the corresponding K_{obs} (b) of $\text{REM}_{7\mu\text{m}}$ and $\text{REM}_{105\mu\text{m}}$. **c** Schematic diagram of the oxidation of $\text{Fe}(\text{CN})_6^{4-}$ on REMs at different fluxes. The convection limit shown in panel a represents the situation where all the influent $\text{Fe}(\text{CN})_6^{4-}$ ions are oxidized. Highlighted areas in panel b correspond to the respective contributions of specific surface area and pore diffusion. Experiments were conducted at 1.91 V_{RHE} in an electrolyte containing 0.1 mM $\text{Fe}(\text{CN})_6^{4-}$, 0.2 mM $\text{Fe}(\text{CN})_6^{3-}$, and 0.33 M NaClO_4 .”

Line 485 – 496 in the revised manuscript:

“The observed mass transfer rate (K_{obs}) was calculated using equation (1)³⁹:

$$K_{\text{obs}} = \frac{I}{zFAC_b} \quad (1)$$

where I is the oxidation current of $\text{Fe}(\text{CN})_6^{4-}$ (A), z represents the number of electrons transferred (1 for the oxidation of $\text{Fe}(\text{CN})_6^{4-}$), F denotes the Faraday constant (96,500 C mol⁻¹), A is the geometry surface area of the electrode (2.54×10^{-4} m²), and C_b is the bulk concentration of $\text{Fe}(\text{CN})_6^{4-}$ (0.1 mol m⁻³).

The volumetrically averaged mass transfer coefficient ($k_m A_e$) was calculated using equation (2)⁴²:

$$k_m A_e = \frac{I}{zFV_e C_b} = \frac{K_{obs}}{L} \quad (2)$$

where k_m is the mass transfer coefficient (m s^{-1}), A_e denotes the active electrode area per unit volume (m^{-1}), L represents the thickness of the porous electrode (m), and V_e is the total volume of the electrode within the reactor ($7.63 \times 10^{-7} \text{ m}^3$)."

Page 19 in the revised Supplementary Information:

“

Supplementary Fig. 14. Relationship between $k_m A_e$ and flux for various REMs²². The performance of mass transfer on REM_{105μm} is superior to reticulated vitreous carbon (RVC) and expanded metal mesh. Further reducing the pore size to 7 μm resulted in improved mass transfer capability that is higher than the carbon fiber materials.”

Comment 7: Line 125: One assumption here that the authors did not mention is that the surface area is linearly correlated to the current increase. Is this assumption valid? Any reference for this assumption? Hence, it is unsure in Figure 2b if the 4 times increase in surface area can cause a 4 times increase in current density.

We thank the reviewer for this important comment. In the previous manuscript, it was imprecise in our previous manuscript to claim that the surface area would consistently lead to a 4-fold increase in oxidation current. Although, according to equation (2), the observed mass transfer rate K_{obs} is proportional to surface area A_e , the

electrochemically active surface area in the REM system does not always equate to the BET surface area. On one hand, the area is affected by nonuniform potential distribution, resulting in a lower fraction of BET area being electrochemically active in smaller pores compared to larger pores. On the other hand, as you mentioned in Comment 6, the enhancement from surface area is correlated with mass transfer, which in turn reduces the overall impact of the increased area. For example, at relatively low fluxes, smaller pores can fully oxidize the $\text{Fe}(\text{CN})_6^{4-}$ while larger pores cannot, decreasing the effective area of smaller pores.

While it is difficult to quantify the exact contributions of surface area and mass transfer, we can roughly separate the kinetics enhancement into two components: (i) the maximum enhancement attributable to surface area (i.e., 4-fold) and (ii) the minimum enhancement attributable to mass transfer, as depicted in Fig. 2b. We also revised the relevant statement and added a schematic (Fig. 2c) in the revised manuscript.

Line 134 – 182 in the revised manuscript:

“To gain insights into the mass transfer mechanism of reactants toward the electrode surface, the oxidation current of $\text{Fe}(\text{CN})_6^{4-}$, a model reactant with high intrinsic electrochemical reactivity, was measured under different fluxes³⁹. To exclude the impact of charge transfer limitation, a relatively low concentration of 0.1 mM $\text{Fe}(\text{CN})_6^{4-}$ was employed for the electrochemical reaction (Supplementary Figs. 12 and 13)⁴⁰. At relatively low membrane flux (e.g., flux $< 0.3 \times 10^{-2} \text{ m s}^{-1}$), the current on $\text{REM}_{7\mu\text{m}}$ linearly changed with flux and was nearly identical to the convection limit³⁹ (Fig. 2a). This indicates that almost all the $\text{Fe}(\text{CN})_6^{4-}$ ions that traverse the membrane were oxidized and the reaction was limited by the convection process³⁹ (Fig. 2c). In contrast, the current observed on $\text{REM}_{105\mu\text{m}}$ was significantly lower than that on $\text{REM}_{7\mu\text{m}}$, indicating the slower mass transfer of $\text{Fe}(\text{CN})_6^{4-}$ within relatively large channels (Fig. 2c). As the flux increased, the anodic current gradually increased, indicating the alleviated concentration polarization of $\text{Fe}(\text{CN})_6^{4-}$ (Fig. 2a and 2c). As the flux continuously increased, the current on $\text{REM}_{7\mu\text{m}}$ gradually deviated from the linear region, suggesting that the reaction is also controlled by the diffusion processes of the reactant molecules⁴¹.

To quantitatively investigate the mass transfer process, the observed mass transfer rate (K_{obs}) was determined from the anodic current according to equation (1)³⁹(Fig. 2b). The mass transfer rate can also be expressed as the product of mass transfer coefficient (k_m) and surface area (A_e), which is widely employed and shown in equation (2)⁴². However, in the REM system, the increase in surface area does not result in a proportional

improvement in kinetics, as validated by the current curves in Fig. 1c. Two factors can contribute to this. First, the potential distribution within the microchannels is nonuniform, influencing the effective reaction area, and it is discussed in detail below. Second, there is a strong interaction between mass transfer and effective surface area, where one factor substantially affects the other. For instance, the influent $\text{Fe}(\text{CN})_6^{4-}$ ions in $\text{REM}_{7\mu\text{m}}$ can be completely oxidized at a relatively low flux, whereas in the $\text{REM}_{105\mu\text{m}}$ at the identical flux, they cannot be fully oxidized (Fig. 2c). Therefore, the effective surface area of $\text{REM}_{7\mu\text{m}}$ decreases.

Although quantifying the contribution of surface area and mass transfer is challenging, we can roughly categorize the kinetics enhancement into two parts: (i) the maximum A_e contribution (i.e., 4-fold) and (ii) the minimum k_m contribution (Fig. 2b). At fluxes $> 0.5 \times 10^{-2} \text{ m s}^{-1}$, $\text{REM}_{7\mu\text{m}}$ exhibited approximately 7-fold higher K_{obs} than $\text{REM}_{105\mu\text{m}}$. Considering that the surface area of $\text{REM}_{7\mu\text{m}}$ is only 4-fold greater than that of $\text{REM}_{105\mu\text{m}}$ (Supplementary Table 1), the enhanced K_{obs} value of $\text{REM}_{7\mu\text{m}}$ cannot be merely attributed to the relatively large surface area. The accelerated mass diffusion contributed by smaller pores is another important reason. Supplementary Fig. 14 depicts the relationship between $k_m A_e$ and flux for various REMs, where the electrode performance from literature was also presented for comparison. $\text{REM}_{7\mu\text{m}}$ is among the best-performing electrodes with high mass transfer rate and reaction kinetics. This enables rapid contaminant removal and chemical transformation.

Fig. 2 Mass transfer in REMs. a, b Oxidation current of $\text{Fe}(\text{CN})_6^{4-}$ (a) on REMs at

different fluxes and the corresponding K_{obs} (b) of REM_{7μm} and REM_{105μm}. c Schematic diagram of the oxidation of Fe(CN)₆⁴⁻ on REMs at different fluxes. The convection limit shown in panel a represents the situation where all the influent Fe(CN)₆⁴⁻ ions are oxidized. Highlighted areas in panel b correspond to the respective contributions of specific surface area and pore diffusion. Experiments were conducted at 1.91 V_{RHE} in an electrolyte containing 0.1 mM Fe(CN)₆⁴⁻, 0.2 mM Fe(CN)₆³⁻, and 0.33 M NaClO₄.”

Comment 8: Lines 166-171: In the text of Energy Consumption Calculation, the current is 50 mA. However, in Fig. 3, the current density is 19.7 mA/cm² (i.e., 1.25 cm * 1.25 cm * π * 19.7 mA/cm² = 97 mA). These two values are different. Which number was used to calculate energy consumption?

We thank the reviewer for this suggestion. In all the electrochemical oxidation experiments in the previous manuscript, a silicone gasket was used to separate the anode from the cathode (Supplementary Fig. 1b). The gasket possesses a circular hole with a diameter of 18 mm to allow for water flow. Although the diameter of the REM electrodes is 25 mm, their edges were covered with the gasket. Therefore, the effective area of the electrode is 0.9 cm * 0.9 cm * π = 2.54 cm². That is, the applied current is 50 mA (i.e., 2.54 cm² * 19.7 mA cm⁻² = 50 mA). Therefore, 50 mA is used to calculate the energy consumption. We have added the relevant statements in the revised manuscript and the “Energy Consumption Calculation” section of the Supplementary Information.

Line 287 – 289 in the revised manuscript:

“... The initial concentration of organics: 100 μM. Current: 19.7 mA cm⁻². Reaction area: 2.54 cm². pH = 7. Error bars represent the data from duplicate tests.”

Line 498 – 501 in the revised manuscript:

“The oxidation experiments of organics were conducted on the reactor similar to Fe(CN)₆⁴⁻ oxidation, where a RuO₂/Ti mesh was used as the cathode. The reaction area, 2.54 cm² (i.e., 0.9 cm × 0.9 cm × π = 2.54 cm²) was defined by a silicone gasket (thickness: 1 mm).”

Line 58 – 61 in the revised Supplementary Information:

“

$$E_E = 10^{-3} \times \frac{V_{cell} I}{Q} \quad (2)$$

where V_{cell} is the cell potential (V), I is the current used in the experiment (i.e., $2.54 \text{ cm}^2 \times 19.7 \text{ mA cm}^{-2} \times 10^{-3} = 0.05 \text{ A}$), Q is the volumetric flow rate at which 90% 4-CP removal was achieved ($\text{m}^3 \text{ h}^{-1}$).

Comment 9: SI Line 230: what is the role of the dash line in the figure?

We thank the reviewer for this suggestion. The dash line here is only used to illustrate the trend of how the enhancement factors change with the DET contributing. In fact, the dash line didn't reveal any mathematical relationship between these two factors. To avoid ambiguity, we have replaced the dash line with an arrow indicating the trend.

Page 28 in the revised Supplementary Information:

“

Supplementary Fig. 23. Relationship between enhancement factors and the DET contribution in REM_{7 μ m} of different model reactants.”

The same confusing line can be found in Fig. 3c, so we also replaced the dash line in it.

Line 281 – 289 in the revised manuscript:

“

Fig. 3 Performance of electrochemical oxidation of organics on REMs. ... c Relationship between enhancement factors and the contribution of DET in REM_{105 μm} for different model reactants.”

Comment 10: It is suggested to add the unit for parameters in the description for all equations in the main text and the SI to increase readability.

We appreciate your important comments which will significantly increase the readability of our manuscript. We have added the unit of parameters for all equations in the main text and the SI.

Line 485 – 496 in the revised manuscript:

“The observed mass transfer rate (K_{obs}) was calculated using equation (1)³⁹:

$$K_{\text{obs}} = \frac{I}{zFAC_b} \quad (1)$$

where I is the oxidation current of $\text{Fe}(\text{CN})_6^{4-}$ (A), z represents the number of electrons transferred (1 for the oxidation of $\text{Fe}(\text{CN})_6^{4-}$), F denotes the Faraday constant ($96,500 \text{ C mol}^{-1}$), A is the geometry surface area of the electrode ($2.54 \times 10^{-4} \text{ m}^2$), and C_b is the bulk concentration of $\text{Fe}(\text{CN})_6^{4-}$ (0.1 mol m^{-3}).

The volumetrically averaged mass transfer coefficient ($k_m A_e$) was calculated using equation (2)⁴²:

$$k_m A_e = \frac{I}{z F V_e C_b} = \frac{K_{obs}}{L} \quad (2)$$

where k_m is the mass transfer coefficient (m s^{-1}), A_e denotes the active electrode area per unit volume (m^{-1}), L represents the thickness of the porous electrode (m), and V_e is the total volume of the electrode within the reactor ($7.63 \times 10^{-7} \text{ m}^3$)."

Line 48 – 65 in the revised Supplementary Information:

“The mineralization current efficiency (MCE) for 4-CP degradation was calculated according to Supplementary equation (1)¹,

$$MCE = \frac{TOC_0 - TOC_t}{1.2 \times 10^4 x I t} n F V \times 100\% \quad (1)$$

where TOC_0 and TOC_t are the values of total organic carbon (TOC, mg C L^{-1}) at the beginning and time t , respectively. x is the number of carbon atoms in a 4-CP molecule, I is the applied current (A), n is the theoretical number of electrons for complete mineralization (estimate to be 26 for 4-CP), F is the Faraday constant ($96,485 \text{ C mol}^{-1}$), and V is the volume of the solution (L).

The total energy consumption was calculated as the sum of electrical energy (E_E , kWh m^{-3}) and pumping energy (E_P) normalized per log removal of 4-CP. The E_E value was calculated according to Supplementary equation (2)²,

$$E_E = 10^{-3} \times \frac{V_{cell} I}{Q} \quad (2)$$

where V_{cell} is the cell potential (V), I is the current used in the experiment (i.e., $2.54 \text{ cm}^2 \times 19.7 \text{ mA cm}^{-2} \times 10^{-3} = 0.05 \text{ A}$), Q is the volumetric flow rate at which 90% 4-CP removal was achieved ($\text{m}^3 \text{ h}^{-1}$).

The E_P value was calculated by Supplementary equation (3)²,

$$E_p = 3.6 \times 10^{-5} \times \frac{\rho g \Delta P}{\eta} \quad (3)$$

where ρ is the density of water (997 kg m⁻³), g is the gravitational constant (9.81 m s⁻²), ΔP is the transmembrane pressure at a given flow rate (bar), and η is the pump efficiency (assumed as 0.7).”

Line 80 – 107 in the revised Supplementary Information:

“As the current density is influenced by the depth of channel^{4,5}, the “secondary current distribution” module was used to investigate the current distribution ($\mathbf{j}(x)$, A m⁻²) in the flow-through system. The Butler-Volmer equation was used to solve for the current distribution (Supplementary equation (4))⁶,

$$i = F A k^0 [e^{-\alpha f(E-E^0)} - e^{(1-\alpha)f(E-E^0)}] \quad (4)$$

where i is the local current (A), F is the faraday constant (C mol⁻¹), A is the surface area of electrode (m²), k^0 is the standard rate constant (m s⁻¹), α is the transfer coefficient, $f = \frac{F}{RT}$, E^0 is the formal potential (V), and E is the electrode potential (V).

The applied current (i_{app} , A) (Fig. 4b) was determined according to the porosity and current in the experiment (Supplementary equation (5)),

$$i_{app} = I \frac{S}{pA} \quad (5)$$

where I is the current in the experiment (0.05 A unless otherwise mentioned), p is the porosity of REM, A is the geometry surface area of the electrode (2.54×10^{-4} m²), and S (m²) is the cross-section area of the simulated channel.

The inner surface of channels was defined as the reaction surface. For simplicity, the generation of •OH in the region with a potential lower than 2.8 V_{RHE} was ignored^{7,8}. In the region with a potential higher than 2.8 V_{RHE}, the production of •OH ($r_{g,\bullet OH}$, mol m⁻² s⁻¹) is expressed in terms of current density by Supplementary equation (6)^{3,9},

$$r_{g,\bullet OH} = k_{g,\bullet OH} \frac{j(x)}{F} \quad (6)$$

where $k_{g,\bullet OH}$ (dimensionless) was estimated using a TA degradation experiment (Supplementary Fig. 34), $j(x)$ is the local current density ($A m^{-2}$).

Similar to the $\bullet OH$ production, a surface reaction kinetic equation was used to model the direct electron transfer (DET) reaction and the flux of reactants at the electrode surface¹⁰ (Supplementary equation (7)),

$$r_{4-CP,DET} = k_{f,4-CP} N_D c_{4-CP} = k_{4-CP,DET} \frac{j(x)}{F} c_{4-CP} \quad (7)$$

Where $k_{f,4-CP}$ represents the forward rate constant of 4-CP DET reaction, N_D is the number of reaction sites per unit of surface area, and c_{4-CP} is the concentration of 4-CP in the electrode surface. Additionally, $k_{4-CP,DET}$, the rate constant ($m^3 mol^{-1}$), was optimized in the simulation.”

Comment 11: SI Fig. 23: How was the percent yield obtained?

Thanks for this suggestion. Due to the side reactions, the yield of HTA is much lower than the consumption of TA. The percentage yield of HTA is the ratio of HTA production to TA consumption, which was determined by experiment. The percentage yields of HTA obtained on different REMs are slightly different, where the largest pore exhibited the largest yield. This may be attributed to the fact that the oxidation of TA via DET route cannot be neglected, where no HTA is produced in this process (*Environ. Sci. Technol.* **2017**, 51, 2355-2365). This leads to an underestimation on the percent yield of HTA, especially on REM_{7 μ m}. Therefore, the percentage yield derived from REM_{105 μ m} (8.8%) was used in the simulation, which is least affected by DET oxidation. We have added the relevant statements in the supporting information.

Page 33 in the revised Supplementary Information:

“

Supplementary Fig. 28. The percent yield of HTA in the experiment. The percent yield is the ratio of HTA production to TA consumption. The initial concentration of TA: 100 μM . Current: 19.7 mA cm^{-2} . HRT: 2.7 s. The percentage yields of HTA obtained on different REMs are slightly different, where the largest pore exhibited the largest yield. This may be attributed to the fact that the oxidation of TA via DET route cannot be neglected, where no HTA is produced in this process⁷. This leads to an underestimation on the percent yield of HTA, especially on REM_{7 μm} . Therefore, the percentage yield derived from REM_{105 μm} (8.8%) was used in the simulation, which is least affected by DET oxidation.”

Comment 12: Fig. 5d: it is suggested to remove the Taijitu sign.

We appreciate your important suggestion. After careful consideration, we agree with you that the Taijitu sign is unnecessary and may leads to misunderstanding, which has been removed in the revised version.

Line 388 – 397 in the revised manuscript:

“

Fig. 5 Multiphysics simulation results of the microreactors. a Visualized $\bullet\text{OH}$ distribution in the cross-section ($d = 0.1 \text{ mm}$) (top) and the corresponding $\bullet\text{OH}$ concentration as a function of distance from the channel wall (bottom). The inset image shows the magnified $\bullet\text{OH}$ profile and volume ratio of the reaction region. **b** Visualized 4-CP distribution in the cross-section ($d = 0.1 \text{ mm}$) (top) and the corresponding 4-CP concentration as a function of distance from the channel wall (bottom). C_c represents the concentration of 4-CP at the center of the channel. The HRT in the simulation is 9.1 s. **c** Simulated pseudo-first-order kinetic constants of 4-CP degradation. **d** Mechanistic illustration of the spatial confinement effect.”

Comment 13: Line 363: REM characterization should be mentioned in the main text. It is totally fine to keep the details in the SI. However, none is stated in the method

section of the main text.

We thank the reviewer for this suggestion. REM characterization is indeed an important part of the method section, as it allows readers to quickly and fully understand this work. Given that the length of the REM characterization section is short, we have moved it to the method section of the main text.

Line 464 – 474 in the revised manuscript:

“Characterization

The morphology of the prepared electrode was examined using a scanning electron microscope (SEM, Hitachi SU8000). XRD patterns of the electrode were recorded using a D8 advance X-ray diffractometer (Bruker) with Cu K α radiation. XPS spectra were acquired on a Thermo ESCALAB250Xi spectrometer with a monochromated Al K α X-ray source. ESR analysis was conducted using a Bruker A300 spectrometer. The pore size distribution was characterized using the mercury intrusion method (AutoPore IV 9500). The porosity of REMs was determined using Archimedes’ drainage method (Supplementary Table 3). All electrooxidation experiments were performed on a CHI660E electrochemical workstation (CH Instruments). All the electrochemical measurements were conducted on a Gamry Interface 1000 electrochemical workstation.”

Comment 14: Line 377: The authors are suggested to provide details about how the REMs with different pore diameters were fabricated. How was the pore size controlled?

We appreciate your important suggestions. The REMs were fabricated with porous Ti substrates with different pore diameters, which were purchased from Nanjing Shinkai Filter Co., Ltd (China). Specifically, the Ti substrates with different pore sizes were made from Ti particles, which were pressed into discs and then sintered under vacuum condition. The pore size distribution is mainly controlled by the size of the particles. We have updated the relevant statements in the revised manuscript.

Line 449 – 452 in the revised manuscript:

“Porous Ti substrates with a thickness of 3 mm were purchased from Nanjing Shinkai Filter Co., Ltd (China), which were fabricated by compressing Ti particles under high pressure. The pore size of the Ti substrate was controlled by the particle size¹⁴.”

Comment 15: Line 390: It should be “(1)”.

We appreciate your important comments. We have corrected the order number of the equation (1). The modification can be seen in Line 486 of the revised manuscript.

Comment 16: Line 418: DET reaction needs to be clarified. What DET reactions have been considered?

Thank you for pointing out the ambiguity in our statement. The “DET reaction” refers to the direct oxidation of the target organic on the electrode surface, which is different from radical oxidation. To improve readability and avoid misunderstanding, we have changed “DET reaction” to “DET oxidation of the target organic”. The specific revisions are as follows:

Line 521 – 524 in the revised manuscript:

“Four crucial reaction processes including DET oxidation of the target organic, radical oxidation of the target organic, radical oxidation of intermediate products, and self-quenching of •OH were constructed using the “chemistry” module to simulate reactions in confined microchannels.”

REVIEWER #2 (Comments to authors)

General Comment: The authors reported an interesting study on the electron transfer and radical oxidation processes in the microchannels of REM, which are important topics in electrocatalysis and environmental remediation. Both experiments and simulations were conducted to investigate the reaction processes. The authors demonstrate that not only the oxidation kinetics but also the electron transfer mechanism can be largely impacted by the pore diameter. The methodology and results of this study are enlightening and could attract wide attention. The data provided are comprehensive and convincing. I suggest the manuscript be accepted for publication after carefully considering the following points.

Many thanks for your recognition and suggestions on this work. Your insightful comments will greatly help us improve the quality of this paper and have been carefully addressed. A point-by-point response on all issues raised is as follows.

Comment 1: Defective TiO₂ is employed as the electrocatalyst of the REM electrodes. Meanwhile, TiO₂ is an extensively studied photocatalyst and could generate hydroxyl radicals under light irradiation. The impact of light on the electrochemical oxidation processes needs to be clarified.

We appreciate the reviewer's valuable comments. Although TiO₂ can act as a photocatalyst, the large bandgap of TiO₂ determines that it could mainly utilize ultraviolet light. However, the experiments were conducted under room light, where almost no ultraviolet irradiation can be reached. Besides, the reactor is made of polymethyl methacrylate with a frosted outer surface, which could block most of the light irradiation. To fully exclude the possible influence of light, we performed an additional control experiment in darkness by wrapping the reactor with tin foil. The results indicate that there is no observable difference between the reactor under room light and that under darkness. Therefore, the possible impact of light on the electrochemical oxidation processes can be excluded. Accordingly, the following data and discussions have been supplemented in the revised manuscript.

Line 203 – 205 in the revised manuscript:

“The degradation performance of 4-CP on REM_{7μm} under room light and dark conditions did not differ (Supplementary Fig. 16). This further excludes the influence of light on the electrooxidation reactions.”

Page 21 in the revised Supplementary Information:

“**Supplementary Fig. 16.** Degradation performance of 4-CP on REM_{7µm} under room light and dark conditions. Current: 19.7 mA cm⁻². Reaction area: 2.54 cm². pH = 7. Error bars represent the data from duplicate tests.”

Comment 2: The authors employed 200 mM TBA to quench •OH radicals and determine the contribution of different electron transfer routes. It's suggested to provide the reaction kinetics at different concentrations of TBA to prove that 200 mM TBA is high enough to fully quench the radicals.

We appreciate your important comments. It is generally accepted that when the quencher concentration is over 100 times that of the target reactant, radical oxidation can be adequately quenched. Therefore, a relatively high concentration of TBA was used. We have supplemented data on the quenching effect at different TBA concentrations as suggested. It can be observed that even at low concentrations (such as 10 mM), TBA still demonstrates a good quenching effect. Further increasing the TBA concentration can improve the quenching effect to some extent. When the TBA concentration reaches 200 mM, the quenching effect ceases to improve. Therefore, 200 mM TBA was used to ensure complete quenching of the hydroxyl radicals.

We have added the relevant statements and figures in the revised manuscript and supplementary information.

Line 228 – 232 in the revised manuscript:

“Although the 4-CP degradation kinetics can be significantly reduced in the presence of 10 mM TBA, further increasing TBA concentration can enhance the quenching effect to some extent (Supplementary Fig. 20). Ultimately, 200 mM TBA was utilized to ensure complete quenching.”

Page 25 in the revised Supplementary Information:

“

Supplementary Fig. 20. The effluent 4-CP concentration as a function of HRT at different TBA concentrations. (a) The effluent 4-CP concentration as a function of HRT on REM_{7µm} and (b) the corresponding pseudo-first-order kinetic constants. (c) The effluent 4-CP concentration as a function of HRT on REM_{105µm} and (d) the corresponding pseudo-first-order kinetic constants. The dash lines represent the fitted degradation curves. The initial concentration of organics: 100 µM. Current: 19.7 mA cm⁻². Error bars represent the data from duplicate tests.”

Comment 3: *The generation of •OH radicals is detected by using TA and coumarin as the probing molecules, respectively. Although these two methods are approved, the*

DMPO trapping method is more acceptable. I suggest the authors provide the ESR signals of DMPO-OH on different REMs.

We appreciate your important comments. Electron spin resonance (ESR) is a direct, specific, and reliable method to detect radicals. Although ESR does not enable absolute quantitation of the •OH concentration, it can determine the relative differences in •OH production between REMs. We performed ESR measurements on REMs with different diameters. At a fixed HRT of 2.7 s, the intensity of the DMPO-OH increased with the pore size of the REMs, which is consistent with the result of the probe test. Therefore, the result of the ESR measurements reinforces the conclusion that the contribution of radical oxidation decreases with the increase of the pore size.

The relevant data and statement in the revised manuscript and supplementary information have been updated.

Line 223 – 226 in the revised manuscript:

“Further, the signal of DMPO-•OH is lower on REMs with smaller pore sizes. This result contrasts with the fact that REM_{7μm} exhibited optimal electrooxidation performance (Fig. 3a). This discrepancy implies the presence of distinct reaction mechanisms in REMs with different pore sizes.”

Page 24 in the revised Supplementary Information:

“

Supplementary Fig. 19. DMPO spin trapping ESR spectra of REMs. (a) The ESR spectra of the REMs. (b) Peak intensity for DMPO-•OH on the REMs. The experiments were performed in the same conditions as the degradation experiment, except that 4-CP

was absent in the electrolyte. The quartet-signal with the intensity of 1:2:2:1 can be attributed to DMPO-•OH²³. The experiments were performed at an HRT of 2.7 s.”

Comment 4: Besides the removal of specific pollutant molecules, the complete oxidation of organics to CO₂ is also critical for water purification. The representative total organic carbon (TOC) values and the corresponding current efficiencies for organic oxidation are suggested to be provided.

We appreciate your professional comments. The effluent TOC values at different HRTs were tested and obvious differences can be observed on different REMs. In comparison to the REM_{105μm}, the REM with smaller pores exhibited much higher TOC removal efficiencies. At an HRT of 54.5 s, the TOC removal rate on REM_{7μm} was 96%, while only 64% of TOC was removed on REM_{105μm}. The result is consistent with the degradation performance of 4-CP on REMs. The corresponding mineralization current efficiency was also calculated. A current efficiency as high as 56% can be achieved on REM_{7μm} at an HRT of 6.1 s, which is 3 times that on REM_{105μm} (19%). The result demonstrates the great advantage of REM_{7μm} in water purification. We have added the relevant statements in the revised manuscript and supplementary information.

Line 206 – 212 in the revised manuscript:

“To further assess the application potential of REMs in pollutant mineralization, the total organic carbon (TOC) values were measured. At an HRT of 54.5 s, the TOC removal rate on REM_{7μm} was 96%, whereas only 64% of TOC was removed on REM_{105μm} (Supplementary Fig. 17a). The corresponding mineralization current efficiency (MCE) was then calculated based on Supplementary equation (1)⁴⁴ (Supplementary Fig. 17b). REM_{7μm} can achieve a current efficiency of 56% at an HRT of 6.1s, which is three times that of REM_{105μm} (19%).”

Line 48 – 54 in the revised Supplementary Information:

“The mineralization current efficiency (MCE) for 4-CP degradation was calculated according to Supplementary equation (1)¹,

$$MCE = \frac{TOC_0 - TOC_t}{1.2 \times 10^4 \times It} nFV \times 100\% \quad (1)$$

where TOC_0 and TOC_t are the values of total organic carbon (TOC, mg C L⁻¹) at the

beginning and time t , respectively. x is the number of carbon atoms in a 4-CP molecule, I is the applied current (A), n is the theoretical number of electrons for complete mineralization (estimate to be 26 for 4-CP), F is the Faraday constant ($96,485 \text{ C mol}^{-1}$), and V is the volume of the solution (L).”

Page 22 in the revised Supplementary Information:

“**Supplementary Fig. 17.** TOC removal performance and corresponding MCE values on REMs at different HRTs. (a) The TOC removal performance on REMs at different HRTs and (b) the corresponding MCE. MCE is calculated assuming that 4-CP is mineralization to CO_2 and H_2O .”

Comment 5: Both the experiments and simulations demonstrate that the oxidation of 4-CP can be drastically enhanced by reducing the pore sizes of REM. However, the lowest pore diameter investigated in this study is $7 \mu\text{m}$, which is not low enough. Although further reducing the pore diameter may be not easy in the experiment, I suppose it is feasible by simulation. This work would be more attractive if the authors could “predict” what would happen when the pore diameter is further reduced.

We thank the reviewer for the constructive suggestions. It is interesting and very meaningful to analyze what will happen when the pore size is further reduced by simulation. Some studies have reported that the degradation performance on REMs would not keep increasing when the pore size is further reduced. Yang (*Electrochim. Acta.* **2020**, *335*, 135634) found that the diffusion process has been greatly enhanced in the large pores ($\sim 15 \mu\text{m}$) and it is difficult to further improve the kinetic reactivity of REMs by reducing the pore size. Chen (*Environ. Int.* **2020**, *140*, 105813) reported that the performance of MP-Ti-ENTA/ SnO_2 - Sb_2O_3 electrodes with a pore size of $151.9 \mu\text{m}$

was better than the electrode with a pore size of 28.4 μm due to the possible blockage of the smaller pores.

In our work, we found that although the DET reaction was enhanced by reducing the pore size, the generation of hydroxyl radicals was impaired to some extent. The combined effect of these two factors makes it a complex system. To investigate the performance of REM at a smaller pore size than 7 μm , we conducted a simulation on channels with a diameter of 4 μm (Supplementary Fig. 33). Due to the absence of the porosity of the “real” REM_{4 μm} , a value of 14% (the same as REM_{7 μm}) was applied. The result shows that the electrooxidation performance of Channel_{4 μm} is slightly inferior to that of Channel_{7 μm} . The result is attributed to the following reasons. (1) The mass transfer efficiency has been significantly enhanced in REM_{7 μm} and the concentration polarization has already been greatly alleviated (Supplementary Fig. 31). Therefore, it is difficult to further improve the mass transfer by reducing the pore size. (2) With the decrease of the pore size, the surface potential was more nonuniformly distributed. The relative area of REMs that is capable of generating $\bullet\text{OH}$ also decreased. Interestingly, reducing the porosity of the model could evidently increase the reaction kinetics. Although the surface area decreases at relatively low porosity, the surface potential increases, thereby more $\bullet\text{OH}$ could be generated.

In a word, although the electrooxidation reactions in the confined channels are rather complex, the methodology developed in this work provides us an opportunity to gain insights into the processes.

We performed the simulation and added the relevant statements in the discussion section.

Line 419 – 425 in the revised manuscript:

“Additionally, the simulation also revealed that the electrooxidation performance might not continue to improve if the pore size is further reduced due to the inhibition of $\bullet\text{OH}$ generation (Supplementary Fig. 33). Furthermore, increasing the mass transfer efficiency becomes challenging as the concentration polarization has already been significantly alleviated in REM_{7 μm} (Supplementary Fig. 31). These findings suggest that there is an optimal pore size for REMs, which is consistent with the results of previous studies^{44,61}.”

Page 38 in the revised Supplementary Information:

“

Supplementary Fig. 33. Simulated pseudo-first-order kinetic constants of 4-CP degradation. The simulation of the channel_{4μm} was performed with 30% (4.2%) and 100% (14%) of the porosity of the REM_{7μm}. The electrooxidation performance did not improve when the channel diameter further decreased to 4 μm, due to the fully alleviated concentration polarization. Nonuniform surface potential distribution and the less reactive area also contribute to the slightly lower performance versus channel_{7μm}. Interestingly, reducing the porosity of the model could evidently increase the reaction kinetics. Although the surface area decreases at relatively low porosity, the surface potential increases, thereby allowing more •OH to be generated.”

Comment 6: *Some of the important parameters for the simulation of HTA generation (Fig. 4c) are not shown, such as the diffusion coefficient of TA, the reaction kinetic constant of TA and OH radical.*

We appreciate your important comments. We supplemented the relevant parameters (the diffusion coefficient of TA, the reaction kinetic constant of TA and •OH, and the HRT for TA oxidation) in Supplementary Table 4.

Page 43 – 44 in the revised Supplementary Information:

“Supplementary Table 4. Parameters and values used in the simulation.

Parameters	Value
Oxygen evolution potential ^a	2.1 [V]

Exchange current density of oxygen evolution	$4 \times 10^{-2} [\text{A m}^{-2}]^{25}$
Electron transferred coefficient for oxygen evolution ^b	0.15
Conductivity of electrolyte	2.54 [S m ⁻¹]
Applied current of one channel ^c	0.059, 0.278, 1.203, 5.333 [μA]
Inlet concentration	$1 \times 10^{-4} [\text{M}]$
HRT for 4-CP simulation	2.7 – 54.5 [s]
HRT for TA simulation	2.7 [s]
$k_{g, \bullet\text{OH}}^{\text{d}}$	8.91×10^{-6}
$k_{4\text{-CP}, \bullet\text{OH}}$	$7.6 \times 10^6 [\text{m}^3 \text{s}^{-1} \text{mol}^{-1}]^{26}$
$k_{\text{TA}, \bullet\text{OH}}$	$4.4 \times 10^6 [\text{m}^3 \text{s}^{-1} \text{mol}^{-1}]^{22}$
$k_{4\text{-CP}, \text{DET}}^{\text{e}}$	0.386 [m ³ mol ⁻¹]
$k_{\bullet\text{OH}, \bullet\text{OH}}$	$5.5 \times 10^6 [\text{m}^3 \text{s}^{-1} \text{mol}^{-1}]^{27}$
$k_{4\text{-CP products}, \bullet\text{OH}}^{\text{f}}$	$5.0 \times 10^6 [\text{m}^3 \text{s}^{-1} \text{mol}^{-1}]$
•OH diffusion coefficient	$2.2 \times 10^{-9} [\text{m}^2 \text{s}^{-1}]^6$
4-CP diffusion coefficient	$4.5 \times 10^{-10} [\text{m}^2 \text{s}^{-1}]^{28}$
TA diffusion coefficient ^g	$0.8 \times 10^{-9} [\text{m}^2 \text{s}^{-1}]$
Tortuosity (τ) ^h	1.7 ⁱ

^aThe oxygen evolution potential was determined based on the experiment (Fig. 1c).

^bThe charge transfer coefficient (α) was determined according to other studies. Although a value of $\alpha = 0.5$ is expected on inactive electrodes, several experimental studies have reported $\alpha < 0.5$ for the OER reaction^{2,31}, even as low as 0.10 for Ti₄O₇². A value of $\alpha = 0.15$ is used in the simulation.

^cThe applied current in representative channels was determined according to the geometric area of electrodes (Supplementary equation (5)).

^dThe rate constant for •OH production was determined based on the experiment (Supplementary Fig. 28).

^eThe rate constant for DET reaction of 4-CP was determined based on the parameters optimization in the simulation.

^fThe rate constant for radical oxidation of intermediate product was determined based on previous studies and parameters optimization in the simulation²⁷.

^gThe diffusion coefficient of TA was estimated according to other organic molecules (0.67 – 0.99 m² s⁻¹)³².

^hThe diffusion coefficient was corrected as $\frac{D}{\tau^2}$ ³³, because the tortuosity of the pores may inhibit the diffusion process of molecules³⁴.

ⁱThe tortuosity was estimated according to a previous study (1.2 – 1.8)³⁵.”

Comment 7: As the current methodology is sophisticated, I think its application to the cathodic systems should be feasible. A brief discussion on such a possibility can be helpful.

We appreciate your important comments. Thanks for your approval of our research methodology. Similar to electrocatalytic oxidation, the reaction pathway and kinetics in cathodic systems are affected by mass transfer, atomic hydrogen (H*) generation, and potential distribution in spatially confined pores. Applying such a methodology to the cathodic systems is very attractive, as electrocatalytic reduction is an emerging technique for wastewater remediation and resource recovery. We have added some statements in the discussion section.

Line 441 – 443 in the revised manuscript:

“Given that electrocatalytic reduction is an emerging technique for wastewater remediation and resource recovery, applying such a methodology to the cathodic systems is another important area of research. In addition, ...”

Comment 8: There are some format errors in this manuscript. For instance, the sequence number of the equation in line 390 should be 1; the unit of the x-axis in Fig. S12a seems to be wrong. The authors should carefully check the whole manuscript to avoid such errors.

We appreciate your important comments. We are sorry for the format errors in the previous manuscript. We have corrected the sequence number of the equations. Combining Comment 2 of reviewer 3, we changed the unit of the x-axis in Fig. S12a (Supplementary Fig.13 in the revised version) to m s^{-1} . We have checked the revised manuscript with the help of an English native speaker to avoid any grammar errors and ambiguous expressions.

REVIEWER #3 (Comments to authors)

General Comment: The manuscript entitled “Unveiling the spatially confined oxidation processes in reactive electrochemical membranes” provides interesting data and the authors found that both the reaction kinetics and the electron transfer pathway can be effectively regulated by the spatial confinement effect. The reviewers believe that the manuscript has positive implications for the field of REM wastewater treatment. Here are my comments:

We thank you very much for your insightful comments on our work. The questions and suggestions raised by you are important and helpful. They make us carefully reflect and will greatly help us improve the quality of our work in the revision. A point-by-point response on all issues raised is as follows.

Comment 1: Microfluidic has some unique fluid properties, such as laminar flow and droplets. I do not really agree that REM can be considered as a microfluidic platform.

We appreciate your important comments. We are sorry for the misuse of the terminology. Microfluidics is the science of manipulating minute amounts of fluids in networks of tiny channels (*Chem. Rev.* **2022**, 122, 7236-7266). The REM cannot be considered a microfluidic platform due to its complex structure. It also cannot precisely manipulate the properties of the fluid. We have removed the relevant statements in the revised manuscript. The revised statements are as follows.

Line 37 – 39 in the revised manuscript:

“To address this problem, reactive electrochemical membranes (REMs), or flow-through electrodes, have been recently developed as a promising solution (Supplementary Fig. 1a)⁹⁻¹¹.”

Line 48 – 49 in the revised manuscript:

“The gap arises from the complex integration of mass transfer and electrochemical reactions in the spatially confined pores.”

Comment 2: Change mL cm⁻² s⁻¹ to m s⁻¹

We appreciate your important comments. In general, $\text{mL cm}^{-2} \text{s}^{-1}$ is often employed as the unit of flux for the researchers in membrane filtration field. However, the unit may be not familiar to readers in other fields. As you said, m s^{-1} is commonly used in figures about mass transport rate and flux. Therefore, we have changed the unit of the x-axis into m s^{-1} in Fig. 2 and Fig. S13.

Line 175 – 182 in the revised manuscript:

“

Fig. 2 Mass transfer in REMs. a, b Oxidation current of Fe(CN)_6^{4-} (a) on REMs at different fluxes and the corresponding K_{obs} (b) of $\text{REM}_{7\mu\text{m}}$ and $\text{REM}_{105\mu\text{m}}$. **c** Schematic diagram of the oxidation of Fe(CN)_6^{4-} on REMs at different fluxes. The convection limit shown in panel a represents the situation where all the influent Fe(CN)_6^{4-} ions are oxidized. Highlighted areas in panel b correspond to the respective contributions of specific surface area and pore diffusion. Experiments were conducted at 1.91 V_{RHE} in an electrolyte containing 0.1 mM Fe(CN)_6^{4-} , 0.2 mM Fe(CN)_6^{3-} , and 0.33 M NaClO_4 .”

Page 18 in the revised Supplementary Information:

“

Supplementary Fig. 13. Oxidation current and charge transfer-limited current of $\text{Fe}(\text{CN})_6^{4-}$ on REMs. (a) Steady-state (mass transfer-limited) oxidation current of $\text{Fe}(\text{CN})_6^{4-}$ on REMs at different fluxes. ...”

Comment 3: Fig. S7, the XPS O1s deconvolution is not reasonable and need to be re-fitted.

We appreciate your professional comments. We have re-fitted it and supplemented the difference between the fitted curve and the raw data. A detailed discussion on the XPS data was also added.

Line 103 – 108 in the revised manuscript:

“After annealing in an argon atmosphere, the oxygen vacancy increased, as shown in the O 1s XPS band (Supplementary Fig. 9a and b)³³. Additionally, the Ti 2p peak and the valence band edge shifted toward lower binding energy (Supplementary Fig. 9c and d), indicating an increased electron density with the introduction of oxygen vacancy, which substantially improves the conductivity and electrochemical activity^{34,35}.”

Page 14 in the revised Supplementary Information:

“

Supplementary Fig. 9. XPS spectra of REM_{7µm} before and after annealing in an argon atmosphere. O 1s XPS spectra of REM_{7µm} before (a) and after (b) annealing in an argon atmosphere. (c) Ti 2p XPS spectra. (d) valence band XPS spectra. For the O 1s spectra, the peaks at 530.3, 531.5, and 532.6 eV were attributed to lattice oxygen (O_L), adsorbed oxygen (O_{ads}), and surface oxygen (H₂O), respectively^{16,17}. The increase of the O_{ads} demonstrates the existence of oxygen vacancies accompanied by localized electrons richness¹⁸. The Ti 2p peak shifted to a lower binding energy by -0.15 eV, indicating the lattice Ti⁴⁺ atoms were partly reduced to Ti³⁺. Consistent with the deconvolution of the O 1s XPS band, the unsaturated Ti³⁺ further suggested the existence of oxygen vacancies¹⁹. The position of the valence band edge shifted from 2.70 eV to 2.50 eV, showing a narrowed band gap after the thermal treatment²⁰.”

Comment 4: *I cannot agree the statement that the REM electrode has high electrochemical stability. As shown in Supplementary Figure 10, the anode potential has increased over 1 V during a 15 h testing at a constant current of 20 mA cm⁻², suggesting that anode oxygen evolution activity has decayed significantly. The reviewer recommends that all REM electrodes should be treated with continuous electrolysis until their activity is stabilized prior to use.*

We thank the reviewer for this important comment. We are sorry for the inappropriate statement. The stability of the electrode is indeed an important factor for evaluating electrode performance. Dimensional stable anodes (DSA) are commonly used in current electrooxidation research. However, there are some limitations when using DSA in mechanistic studies. The major issue is that coatings cannot be uniformly loaded into pores, causing blockage of pores and preventing quantitative determination of active surface area (*Environ. Int.* **2020**, *140*, 105813). This severely affects our research on the reaction mechanism within the pores.

Cai (*Appl. Catal. B.* **2019**, *257*, 117902) and Lim (*Environ. Sci. Technol.* **2019**, *53*, 6972-6980) found that introducing oxygen vacancies into TiO₂ can significantly enhance its conductivity and enable it to serve as an electrocatalyst for organic degradation. We adopted a hydrothermal method to fabricate TiO₂ nanosheets on the Ti substrate. Utilizing the in-situ dissolution-recrystallization process of Ti as a catalyst precursor, TiO₂ was uniformly grown in the pores. This method effectively avoids limitations faced by other coating electrodes and is highly suitable for the mechanistic study in this work.

Specifically, after annealing in the argon atmosphere, all the prepared electrodes needed to be treated with pre-electrolysis for 3 hours. To avoid the influence of the electrode activity and potential change on the electrooxidation performance, we only collected the degradation data in the first 3 hours for analysis. Over the experimental time (180 min), the potential change was only 0.1 V. In addition, the degradation experiment was repeated 3 times within this time period, and the result showed very good stability.

Moreover, we performed duplicate experiments with different electrodes, which showed good repeatability in potential and degradation results. The degradation performance on REMs with different pore sizes was significantly different, as seen in Fig.3a and b, which demonstrate the validity of our experimental results.

Based on the above analysis, although the electrodes we used have lower stability compared to commercial electrodes, they fully meet the requirements for this mechanistic study. The data we obtained also showed great accuracy and repeatability. Our results can guide the preparation and application of practical electrodes.

To improve the preciseness of this manuscript and address the reviewer's concern, we have updated Fig. 11 in Supplementary Information (Fig. S10 in the previous version) and revised the statements in the main text.

Line 123 – 126 in the revised manuscript:

“Moreover, steady potentials and electrooxidation performance were maintained throughout the experimental duration at a fixed current density, underscoring the capability of REMs in serving as model electrodes (Supplementary Fig. 11).”

Page 16 in the revised Supplementary Information:

Supplementary Fig. 11. Potential-time curve and 4-CP degradation performance of REM_{7µm} at a current density of 19.7 mA cm⁻². Electrolyte: 0.33 M NaClO₄. All prepared electrodes were subjected to pre-electrolysis for 3 hours. Subsequently, all experiments on each electrode were performed within 180 minutes after the pre-electrolysis.”

Comment 5: *Fig. 2b and Supplementary Figure 13, since the Fe(CN)₆⁴⁻ electron transfer reaction is carried out under mass transfer control, there is no comparability between the reaction surface area and the reaction current. We cannot say briefly which is the enhanced reaction activity due to the enhancement of reaction area.*

We appreciate your important and valuable comments. As you said, there is no comparability between the specific surface area and the reaction current when the reaction is fully controlled by mass transfer (i.e., the convection towards the electrode surface). In this case, Fe(CN)₆⁴⁻ is completely oxidized and the current depends on the convection rate. With the increase of the flux, Fe(CN)₆⁴⁻ cannot be fully oxidized and all of the surface within the pore can serve as the reaction area. In this case, both surface area and diffusion contribute to the reaction rate on REMs, as demonstrated in the current-potential characteristic (*Electrochemical Methods: Fundamentals and*

Applications (Wiley, 2001)).

$$i = F A k^0 [C_O e^{-\alpha f(E-E^0)} - C_R e^{(1-\alpha)f(E-E^0)}]$$

F is the faraday constant (96485 C mol⁻¹), A is the surface area of electrode (m²), C_O is the surface concentration of the oxidative species (mol m⁻³), C_R is the surface concentration of the reductive species (mol m⁻³), k^0 is the standard rate constant (m s⁻¹), α is the transfer coefficient, $f = \frac{F}{RT}$, E^0 is the formal potential (V), and E is the electrode potential (V).

In this case, it is still challenging to quantitatively distinguish the contributions of surface area and mass transfer enhancement to the reaction kinetics in our system. On one hand, the area is affected by the nonuniform potential distribution, with a lower fraction of the BET area being electrochemically active in smaller pores compared to larger pores. On the other hand, the enhancement from surface area is correlated with mass transfer, which in turn reduces the overall impact of the increased area. For example, at relatively low fluxes, smaller pores can fully oxidize the Fe(CN)₆⁴⁻ while larger pores cannot, decreasing the effective area of smaller pores.

While it is difficult to quantify the precise contributions of surface area and mass transfer, we can roughly separate the kinetics enhancement into two components: (i) the maximum enhancement attributable to surface area (i.e., 4-fold) and (ii) the minimum enhancement attributable to mass transfer, as depicted in the revised Fig. 2b. Combined with Comments 6 and 7 of Reviewer#1, the authors have decided to remove SI Fig. 13 from the revised version due to its repetitive and confusing content.

The revised figure clearly shows that while surface area can enhance the kinetic constant to some extent, the confinement-enhanced diffusion also plays a significant role. Accordingly, the figures and discussions have been updated as follows:

Line 134 – 182 in the revised manuscript:

“To gain insights into the mass transfer mechanism of reactants toward the electrode surface, the oxidation current of Fe(CN)₆⁴⁻, a model reactant with high intrinsic electrochemical reactivity, was measured under different fluxes³⁹. To exclude the impact of charge transfer limitation, a relatively low concentration of 0.1 mM Fe(CN)₆⁴⁻ was employed for the electrochemical reaction (Supplementary Figs. 12 and 13)⁴⁰. At relatively low membrane flux (e.g., flux < 0.3 × 10⁻² m s⁻¹), the current on REM_{7μm} linearly changed with flux and was nearly identical to the convection limit³⁹ (Fig. 2a). This indicates that almost all the Fe(CN)₆⁴⁻ ions that traverse the membrane

were oxidized and the reaction was limited by the convection process³⁹ (Fig. 2c). In contrast, the current observed on REM_{105μm} was significantly lower than that on REM_{7μm}, indicating the slower mass transfer of Fe(CN)₆⁴⁻ within relatively large channels (Fig. 2c). As the flux increased, the anodic current gradually increased, indicating the alleviated concentration polarization of Fe(CN)₆⁴⁻ (Fig. 2a and 2c). As the flux continuously increased, the current on REM_{7μm} gradually deviated from the linear region, suggesting that the reaction is also controlled by the diffusion processes of the reactant molecules⁴¹.

To quantitatively investigate the mass transfer process, the observed mass transfer rate (K_{obs}) was determined from the anodic current according to equation (1)³⁹(Fig. 2b). The mass transfer rate can also be expressed as the product of mass transfer coefficient (k_m) and surface area (A_e), which is widely employed and shown in equation (2)⁴². However, in the REM system, the increase in surface area does not result in a proportional improvement in kinetics, as validated by the current curves in Fig. 1c. Two factors can contribute to this. First, the potential distribution within the microchannels is nonuniform, influencing the effective reaction area, and it is discussed in detail below. Second, there is a strong interaction between mass transfer and effective surface area, where one factor substantially affects the other. For instance, the influent Fe(CN)₆⁴⁻ ions in REM_{7μm} can be completely oxidized at a relatively low flux, whereas in the REM_{105μm} at the identical flux, they cannot be fully oxidized (Fig. 2c). Therefore, the effective surface area of REM_{7μm} decreases.

Although quantifying the contribution of surface area and mass transfer is challenging, we can roughly categorize the kinetics enhancement into two parts: (i) the maximum A_e contribution (i.e., 4-fold) and (ii) the minimum k_m contribution (Fig. 2b). At fluxes $> 0.5 \times 10^{-2} \text{ m s}^{-1}$, REM_{7μm} exhibited approximately 7-fold higher K_{obs} than REM_{105μm}. Considering that the surface area of REM_{7μm} is only 4-fold greater than that of REM_{105μm} (Supplementary Table 1), the enhanced K_{obs} value of REM_{7μm} cannot be merely attributed to the relatively large surface area. The accelerated mass diffusion contributed by smaller pores is another important reason. Supplementary Fig. 14 depicts the relationship between $k_m A_e$ and flux for various REMs, where the electrode performance from literature was also presented for comparison. REM_{7μm} is among the best-performing electrodes with high mass transfer rate and reaction kinetics. This enables rapid contaminant removal and chemical transformation.

Fig. 2 Mass transfer in REMs. a, b Oxidation current of $\text{Fe}(\text{CN})_6^{4-}$ (a) on REMs at different fluxes and the corresponding K_{obs} (b) of $\text{REM}_{7\mu\text{m}}$ and $\text{REM}_{105\mu\text{m}}$. **c** Schematic diagram of the oxidation of $\text{Fe}(\text{CN})_6^{4-}$ on REMs at different fluxes. The convection limit shown in panel a represents the situation where all the influent $\text{Fe}(\text{CN})_6^{4-}$ ions are oxidized. Highlighted areas in panel b correspond to the respective contributions of specific surface area and pore diffusion. Experiments were conducted at 1.91 V_{RHE} in an electrolyte containing 0.1 mM $\text{Fe}(\text{CN})_6^{4-}$, 0.2 mM $\text{Fe}(\text{CN})_6^{3-}$, and 0.33 M NaClO_4 .”

Comment 6: Supplementary Figure 12, how did the authors obtain the limiting currents? Being in the reaction control region, the reaction limiting current should be linearly related to the number of active sites, i.e., the electroactive area of the REM electrode. The authors need to reevaluate the calculation methods they use.

We thank the reviewer for this professional comment. As you pointed out, in principle, the charge-transfer limited current i_k (i.e., the limiting current in the reaction control region) should be proportional to the surface area, which is illustrated by the following equation (*Electrochemical Methods: Fundamentals and Applications* (Wiley, 2001)).

$$i_k = FAcC^*$$

Here, A represents the surface area, k represents the rate constant, and C^* represents the bulk concentration. In principle, the charge-transfer limited current should be measured when the concentration polarization is fully alleviated. Thus, it needs to be determined in the initial stage of the electrochemical reaction. As the reaction proceeds, concentration polarization occurs and the current decrease, leading to inaccurate results.

In the previous experiment, we chose the initial current (0.1 s) of the i - t curve as the charge-transfer limited current (Supplementary Fig. 12 in the revised manuscript, *ACS Nano*. **2019**, *13*, 6998-7009). To minimize the impacts of concentration polarization and sampling error, the sampling interval was set to 0.002 s. The average current of 0.002 – 0.020 s was calculated as the charge-transfer limited current. Using the revised sampling method, the charge-transfer limited current slightly increased.

The current on REM_{7 μ m} was 28.0 mA cm⁻² in the presence of 0.1 mM Fe(CN)₆⁴⁻, nearly 4 times that on REM_{105 μ m} (7.8 mA cm⁻²), showing a good linear relationship with the specific surface area. However, with the use of 1 mM Fe(CN)₆⁴⁻, the charge-transfer limited current on REM_{7 μ m} (48.3 mA cm⁻²) was only 2.8-fold that of REM_{105 μ m} (17.1 Ma cm⁻²). In this case, the current ratio of REM_{7 μ m} to REM_{105 μ m} is lower than the corresponding area ratio.

This can be attributed to the following two reasons. (1) The electrical resistance in the electrochemical system hinders the linear improvement of current. As evidenced by the electrochemical impedance spectroscopy (EIS) measurements, the REMs exhibit a similar series resistance of 4.5 Ω . Although the resistance is relatively low compared to reported values, it still leads to a significant potential drop at a high current. (2) The increased current may exacerbate the nonuniform potential distribution within the pores (Fig. 4), which shows a larger impact on the reactive surface area of REM_{7 μ m} than REM_{105 μ m}.

In summary, a relatively low concentration of 0.1 mM Fe(CN)₆⁴⁻ was employed for the electrochemical reaction to avoid the interference of electrical resistance and nonuniform potential distribution. The average oxidation current at the initial phase (0.002 – 0.020 s) was calculated as the charge-transfer limited current. Meanwhile, the current at a steady state (100 – 120 s) was employed as the mass-transfer limited current in Fig. 2a.

Accordingly, the following data and discussions have been updated in the revised manuscript and Supplementary Information.

Line 120 – 123 in the revised manuscript:

“The Nyquist curves of REMs exhibited low series resistance of ca. 4.5Ω , indicating the high conductivity of Ti substrate (Supplementary Fig. 10)³⁸. Additionally, REM_{7 μ m} exhibited the smallest arc radius, indicating the fastest electron transfer across the electrode-solution interface.”

Page 15 in the revised Supplementary Information:

“**Supplementary Fig. 10.** Nyquist plots of REMs. The data was obtained at 0.61 V_{RHE} in 0.33 M NaClO₄ (pH = 7).”

Page 17 – 18 in the revised Supplementary Information:

“

Supplementary Fig. 12. Determination of oxidation current and charge transfer-limited current. (a) Representative linear sweep voltammetry (LSV) curve of REM_{105μm} for Fe(CN)₆⁴⁻ oxidation. (b) Representative chronoamperometry curves to derive the charge transfer-limited current and steady-state oxidation current of Fe(CN)₆⁴⁻. LSV measurements were conducted at a membrane flux of $0.11 \times 10^{-2} \text{ m s}^{-1}$ in an electrolyte containing 1 mM Fe(CN)₆⁴⁻, 2 mM Fe(CN)₆³⁻, and 0.33 M NaClO₄, which was briefly denoted as 1 mM Fe(CN)₆⁴⁻. A potential of 1.91 V_{RHE} was selected to avoid the side reactions (i.e., oxidation of water molecules). The average oxidation current at the initial phase (0.002 – 0.020 s) was calculated as the charge-transfer limited current. Meanwhile, the current at a steady state (100 – 120 s) was employed as the mass-transfer limited current.”

Supplementary Fig. 13. ... (b) Charge transfer-limited current on REMs at different concentrations of Fe(CN)₆⁴⁻. ... The charge-transfer limited current was determined according to Supplementary Fig. 12b. The charge-transfer limited current was not linearly related to the electroactive area of REMs when 1 mM Fe(CN)₆⁴⁻ was used. This can be attributed to the electrical resistance (Supplementary Fig. 10), and nonuniform potential distribution (Fig. 4b) in the experimental system, which affects the charge transfer in REMs. ...”

Comment 7: Fig. 2, the achieved k_m values (Y-axis, which should actually be k_{obs}) should be compared with the literature.

We thank the reviewer for this constructive suggestion. As you pointed out, the k_m value in the previous manuscript is actually the observed mass transfer rate, which is denoted as K_{obs} in the revised manuscript. Here we did not use the symbol k_{obs} to avoid

confusion with the observed reaction kinetics. According to previous researches related to reactive electrochemical membranes (*ACS Nano* **2019**, 13, 6998-7009; *Proc. Natl. Acad. Sci. U. S. A.* **2021**, 118; *Adv. Funct. Mater.* **2023**, 2214725; *Electrochem. Commun.* **2017**, 77, 133-137), volumetrically averaged mass transfer coefficient ($k_m A_e$) is commonly used to compare the performance of REMs, which is directly related to K_{obs} . The comparison of $k_m A_e$ values with various REMs was shown in Supplementary Fig. 14. The performance of mass transfer on REM_{105μm} is superior to reticulated vitreous carbon (RVC) and expanded metal mesh. Further reducing the pore size to 7 μm resulted in improved mass transfer capability that is higher than the carbon fiber materials.

Accordingly, the following data and discussions have been supplemented in the revised manuscript.

Line 170 – 174 in the revised manuscript:

“Supplementary Fig. 14 depicts the relationship between $k_m A_e$ and flux for various REMs, where the electrode performance from literature was also presented for comparison. REM_{7μm} is among the best-performing electrodes with high mass transfer rate and reaction kinetics. This enables rapid contaminant removal and chemical transformation.”

Line 485 – 496 in the revised manuscript:

“The observed mass transfer rate (K_{obs}) was calculated using equation (1)³⁹:

$$K_{obs} = \frac{I}{zFAC_b} \quad (1)$$

where I is the oxidation current of $\text{Fe}(\text{CN})_6^{4-}$ (A), z represents the number of electrons transferred (1 for the oxidation of $\text{Fe}(\text{CN})_6^{4-}$), F denotes the Faraday constant (96,500 C mol⁻¹), A is the geometry surface area of the electrode (2.54×10^{-4} m²), and C_b is the bulk concentration of $\text{Fe}(\text{CN})_6^{4-}$ (0.1 mol m⁻³).

The volumetrically averaged mass transfer coefficient ($k_m A_e$) was calculated using equation (2)⁴²:

$$k_m A_e = \frac{I}{zFV_e C_b} = \frac{K_{obs}}{L} \quad (2)$$

where k_m is the mass transfer coefficient (m s⁻¹), A_e denotes the active electrode area

per unit volume (m^{-1}), L represents the thickness of the porous electrode (m), and V_e is the total volume of the electrode within the reactor ($7.63 \times 10^{-7} \text{ m}^3$).”

Page 19 in the revised Supplementary Information:

“

Supplementary Fig. 14. Relationship between $k_m A_e$ and flux for various REMs²². The performance of mass transfer on REM_{105µm} is superior to reticulated vitreous carbon (RVC) and expanded metal mesh. Further reducing the pore size to 7 µm resulted in improved mass transfer capability that is higher than the carbon fiber materials.”

Comment 8: Figure 3b clearly shows that the absolute contribution of hydroxyl radicals to 4-CP oxidation on REM_{7µm} exceeds the total amount of 4-CP oxidized by REM_{105µm} (including DET oxidation and hydroxyl radicals mediated oxidation) by a factor of 3, indicating that the hydroxyl radical yield of REM_{7µm} is much higher than that of REM_{105µm}. However, this contradicts many of the data in the manuscript, as shown in Fig. 3c,3d, Supplementary Figures 18-20, etc.

We appreciate your important comments, which promoted us to reevaluate the experimental results. By analyzing the data in the manuscript, we believe that the contradiction can be attributed to the two following reasons.

- (1) The HRT in the previous manuscript was calculated based on the volume and

the porosity of REMs, which is widely used in flow-through systems (*Environ. Sci. Technol.* **2020**, 54, 10868-10875; *Environ. Sci. Technol. Lett.* **2019**, 6, 504-510). Based on this method, lower HRTs would be obtained in REMs with lower porosity at the same flux and current. In this way, the reaction kinetics were overestimated for the REMs with relatively low porosity, i.e., the REMs with smaller pores in our work. Meanwhile, the production of HTA and 7-OH COU was measured at the same flux, unlike the method used to calculate kinetic constants in Fig. 3b. After careful consideration, we realize that such a method of calculating HRTs may be not very suitable for our research. Therefore, we recalculated the HRT based on the volume of the REMs ($0.3 \text{ cm} * \pi * 0.9 \text{ cm} * 0.9 \text{ cm} = 0.76 \text{ cm}^3$) in the revised manuscript, where the porosity was not taken into consideration. The data in Fig. 3, S21, and S22 were recalculated using the revised calculation method. All simulations were also recalculated under the same HRT. Based on the recalculated results, the pseudo-first-order kinetic constant of 4-CP degradation on REM_{7 μ m} was 4.1 times higher than that on REM_{105 μ m}. The absolute contribution of radical oxidation on REM_{7 μ m} (6.57 min^{-1}) was comparable to the total oxidation ability on REM_{105 μ m} (4.57 min^{-1}). In a word, the difference between Fig. 3a and 3d in terms of HRT calculation can partially explain the contradiction, which has been amended in the revised version.

(2) The quenching test and probe test are two different ways to quantify the contribution of radical oxidation, and cannot be directly compared. Quantifying the contribution of radicals is a very popular and important topic in environmental and catalytical fields. Currently, there are three main methods (*Water Res.* **2022**, 217, 118425): (i) compare the kinetics before and after adding the quenchers; (ii) directly quantify the radicals by using the probes; (iii) identify and quantify the radicals by ESR measurements. The mechanisms of these three methods are slightly different, as they quantify the radical contribution in different ways. Therefore, it is normal for the data obtained by these three methods to have some deviations. In most cases, combining multiple methods is needed to reach a reliable conclusion.

Specifically, for method 1, the radical contribution to the kinetic constant may be overestimated (compared with method 2) due to the depletion of the reactant. When reactants are consumed rapidly, diffusion would play a more important role in improving the kinetic constant on REM_{7 μ m} (*Chem. Eng. J.* **2018**, 347, 731-740; *J. Hazard. Mater.* **2006**, 138, 614-619). This idea can be confirmed by the results in Supplementary Fig. 15, as $k_{7\mu\text{m}}/k_{105\mu\text{m}}$ increased with the applied current. In addition, the DET reaction played an important role in the 4-CP degradation experiment, reducing the 4-CP concentration in the reaction system and leading to rapid consumption of 4-CP. This effect would further enhance the absolute contribution of

radical oxidation on REM_{7μm}. In contrast, terephthalic acid (TA, the model reactant in Fig. 3d) is resistant to DET reaction (*Environ. Sci. Technol.* **2017**, 59, 2355-2365). The experiment was performed at a relatively low HRT of 2.7 s, at which TA was barely degraded (Supplementary Fig. 22g), indicating the experiment (Fig. 3d) was less affected by the mass transfer. The probe test is more objective since is less affected by the DET process.

To increase the reliability of the result, we performed ESR measurements on REMs (Supplementary Fig. 19). The result shows the production of •OH on REM_{7μm} is lower than that on REM_{105μm}, aligning well with the HTA production. Since ESR measurement is considered a more direct method to quantify radicals, we believe the result is reasonably accurate.

According to the results of the ESR measurement and probe test (HTA production), the •OH production on REM_{7μm} is indeed lower than that on REM_{105μm}. Despite the lower generation of •OH, the faster mass transfer on REM_{7μm} facilitates radical oxidation, especially when the reactants have been heavily consumed by the DET reaction. Therefore, it is reasonable that the reaction kinetics contributed by •OH oxidation on REM_{7μm} was slightly higher than that on REM_{105μm}, although the •OH production on REM_{7μm} was lower. To improve the readability and rigorousness, the following revisions were made to the manuscript and Supplementary Information.

We have revised the calculation method of HRT in the whole text. Fig. 3, 4, and 5 in the main text, Fig. 21, 22, and 23 in the Supplementary Information, and the relevant statement have been revised. All simulations were also recalculated under the same HRT. The main revisions are shown below, while the details can be found in the revised text.

Line 265 – 273 in the revised manuscript:

“Although the production of HTA on REM_{7μm} is only 22% of that on REM_{105μm}, it is important to note that the contribution of radical oxidation to the 4-CP oxidation on REM_{7μm} is higher than that on REM_{105μm} (Fig. 3b). This can be attributed to the different mechanisms of the two methods (i.e., quenching experiment and probe test). The HTA production experiment was less affected by mass transfer due to the lower HRT and resistance to DET reaction. However, in the quenching experiment, the impact of DET process on the radical reaction cannot be neglected. Specifically, the reactants have been heavily depleted by the DET reaction in smaller pores, which would improve the absolute contribution of radical oxidation^{51,52}.”

Line 281 – 289 in the revised manuscript:

“

Fig. 3 Performance of electrochemical oxidation of organics on REMs. a, b Effluent 4-CP concentration (a) as a function of HRT in different REMs and (b) The corresponding pseudo-first-order kinetic constants of 4-CP degradation. **c** Relationship between enhancement factors and the contribution of DET in $\text{REM}_{105\mu\text{m}}$ for different model reactants. **d** Production of HTA at an HRT of 2.7 s (left axis) and the electrode potential in degradation experiment (right axis) on REMs. The initial concentration of organics: 100 μM . Current: 19.7 mA cm^{-2} . Reaction area: 2.54 cm^2 . pH = 7. Error bars represent the data from duplicate tests.”

Line 348 – 351 in the revised manuscript:

“The larger reaction region in channel $7\mu\text{m}$ suggests that radical oxidation can also benefit from the size reduction, which explains the higher absolute contribution of radical oxidation to the 4-CP oxidation on $\text{REM}_{7\mu\text{m}}$ (Fig. 3b) despite the lower $\bullet\text{OH}$ production (Fig. 3d).”

Comment 9: Many experiments, for which the authors only give data under 20 mA cm⁻². The reviewer suggests that the authors perform more experiments at lower as well as higher current densities (electrode potentials).

We appreciate the reviewer's valuable comments. The mass transfer process and radical generation would be affected by the change in current density. Therefore, we performed 4-CP degradation experiments at current densities of 9.8 mA cm⁻² and 39.3 mA cm⁻² (i.e., 25 mA and 100 mA), respectively. With the increase in current density (from 9.8 mA cm⁻² to 39.3 mA cm⁻²), the enhancement factor ($k_{7\mu\text{m}}/k_{105\mu\text{m}}$) increased from 2.4 to 5.1. The result can be mainly attributed to the intensified concentration polarization at high currents. At a higher current density, the surface reaction rate was increased, which requires faster mass transfer in the electrode pores. Consequently, the kinetic constant increased by nearly 4 times on REM_{7 μm} (from 9.2 min⁻¹ to 36.3 min⁻¹) when increasing the current by four-fold. In comparison, it only increased by less than 2 times on REM_{105 μm} (from 3.9 min⁻¹ to 7.1 min⁻¹). In addition, the increased current raised the surface potential, meaning that REM_{7 μm} could generate more •OH, which is one of the factors limiting its performance.

We have updated the relevant statements and figures in the revised manuscript and supplementary information.

Line 196 – 203 in the revised manuscript:

“The 4-CP degradation experiments at current densities of 9.8 mA cm⁻² and 39.3 mA cm⁻² were also conducted (Supplementary Fig. 15). The degradation performance of REMs was significantly enhanced, especially on REM_{7 μm} , with an increase in the current density. The kinetic constant on REM_{7 μm} (36.3 min⁻¹) was 5.2 times that of REM_{105 μm} (7.1 min⁻¹) at 39.3 mA cm⁻². This can be primarily attributed to the relatively fast mass transfer on REM_{7 μm} , which well matched the elevated current. Moreover, the potential increase at a higher current could facilitate REM_{7 μm} in generating more •OH, which is detailly discussed in the next section.”

Line 373 – 376 in the revised manuscript:

“This simulation of concentration distribution also explains the result in Supplementary Fig. 15. It shows that as the current increases, the reaction on REM_{7 μm} is enhanced to a greater extent than that on REM_{105 μm} .”

Page 20 in the revised Supplementary Information:

“

Supplementary Fig. 15. The 4-CP degradation performance on REMs at current densities of 9.8 mA cm⁻² and 39.3 mA cm⁻². (a) The effluent 4-CP concentration as a function of HRT at a current density of 9.8 mA cm⁻². (b) The effluent 4-CP concentration as a function of HRT at a current density of 39.3 mA cm⁻². (c) The comparison of pseudo-first-order kinetic constants on REMs at different current densities.”

Comment 10: Frankly, the reviewer has doubts about the simulation results of 4-CP degradation kinetics (2.24 s⁻¹ vs 2.25 s⁻¹) and the TA reaction to produce HTA (1.79 μM vs 1.76 μM, Fig. 4b). As we know, REM substrate (porous titanium filter) used in this study is made of sintered titanium particles, which determines that the pore structure inside the REM electrode is very complex and variable, and the pore size distribution is also very heterogeneous (Fig. 1b). Although the assumption of microtubules can simplify the computational model to the maximum extent (There is a huge difference between the model and the actual situation), it is often difficult to give accurate quantitative results. In addition, the bubbles generated by the electrolysis process can also affect the simulation results by perturbing the boundary layer as well as the distribution of hydroxyl radicals. Supplementary Figure 23 shows that the selectivity towards HTA produced from the oxidation of TA under REM with different pore size is also variable (although the authors chose a mean value of 7% for the simulation). What the authors need to explain is how they achieved simulation results (Fig. 4b) that are so close to the accurate experimental results using a distortion model with many simplifications.

We appreciate your important comments. As you said, our simulation was based on a simplified model, in which the highly complex and heterogeneous pore structure was simplified to microchannels. The purpose of the simulation is to analyze the effects of the pore size on the DET reaction and radical oxidation. The surface potential distribution and the radical oxidation region can also be visualized, which helps us to analyze the microscopic reaction mechanisms that can be hardly investigated by experiments. In addition, parameters can be controlled individually in the simplified microchannel model, helping us analyze the role of each factor.

The good fitting of the simulation results can be attributed to the rationally selected parameters. Most of the parameters in the model are derived from the experimental results. Others are also supported by the literature. The main parameters and their selection process are shown as follows.

(1) The pore sizes of the microchannels were set to 7.3, 17.5, 45.0, and 105.0 μm, respectively. These values are consistent with the predominant size that accounted for the largest percentage in the differential mercury intrusion of REMs (Fig. 1b). The initial concentration (100 μM) and the hydraulic residence time (HRT, 2.7 –54.5 s) in the simulation were also determined according to the actual situation. The applied current on each microchannel was calculated based on the porosity and current in the experiment, as shown in Supplementary equation (5). Using this calculation method, the apparent current density in the simulated microreactors was the same and consistent

with the actual situation.

(2) The current and potential distributions were modeled using the well-established Butler-Volmer equation (*Electrochemical Methods: Fundamentals and Applications* (Wiley, 2001)). The onset potential of the oxygen evolution reaction (OER) on simulated microchannels was 2.1 V_{RHE}, which is determined by the experiment (Fig. 1c). The exchange current density (i_0) for the OER reaction on defective TiO₂ was not found in the literature. Therefore, a value for Ti₄O₇ (4×10^{-3} mA cm⁻², *Electrochim. Acta.* **2016**, 214, 326-335) was used. The charge transfer coefficient (α) was determined according to other studies. Although a value of $\alpha = 0.5$ is expected on inactive electrodes, several experimental studies have reported $\alpha < 0.5$ for the OER reaction (*ACS ES. T. Eng.* **2022**, 2, 713-725; *Electrochem. Commun.* **2008**, 10, 607-610), even as low as 0.10 for Ti₄O₇ (*ACS ES. T. Eng.* **2022**, 2, 713-725). A value of $\alpha = 0.15$ is used in the simulation. The conductivity of the electrolyte (2.56 S m⁻¹) was determined according to the experiment condition. The charge transfer coefficient and exchange current density are two critical parameters that influence the surface potential distribution. With the increase of i_0 and α , the surface potential decreases. However, the phenomenon that the surface potential was more nonuniformly distributed in the small channels would not change.

(3) The kinetic constants and the diffusion coefficient were determined according to other studies (Supplementary Table 4). Specifically, the •OH generation rate was estimated using a TA degradation experiment on the plate electrode (Supplementary Fig. 34). In the simulation of HTA production, the selectivity towards HTA was set to 8.8% in the revised manuscript, which is the selectivity on REM_{105μm} in the experiment. It should be noted that only a proportion of TA can be selectively oxidized to HTA due to side reactions and overoxidation of HTA (*J. Environ. Monit.* **2010**, 12, 1658-1665). In addition, TA could also be slowly oxidized by DET process and HTA cannot be produced in this process. This can explain the fact that the percent yield was lower on REM_{7μm}. To better simulate the •OH production in the simulation, and minimize the influence of DET oxidation on the percent yield, a value of 8.8% was used in the simulation.

(4) The direct oxidation process of 4-CP was simulated by Supplementary equation (7), where the DET process was simplified to a surface reaction process. The number of reaction sites is assumed to be proportional to the current density, so the DET reaction rate is proportional to $k_{4-CP,DET}$, local current density, and the surface concentration of the reactant. The $k_{4-CP,DET}$ value is influenced by factors like reactant and catalyst species, so the value could not be obtained from other literature. Therefore, the $k_{4-CP,DET}$

value was optimized in the simulation based on the experimental result at the applied current density of 19.7 mA cm^{-2} . Using this optimal parameter, the simulation of the degradation at different current densities was also performed (Fig. 5c). The effectiveness of our model and parameters can be evidenced by the continued good agreement between the simulation results and the experimental data.

Other simulation parameters are detailed in the Supplementary Information, along with explanations of their references and rationales for selection.

The pore structure and the bubbles are two important issues that we did not take into account in the model. The pore size used in the simulation is the predominant pore size in the fabricated electrodes. Although the pores were irregularly structured, their pore size distribution was relatively concentrated. As shown in Supplementary Fig. 4, over 45% of the total pore volume falls in the $\pm 40\%$ range around the predominant size. In addition, although the pore structure may have an influence on the mass transfer process, this effect can be taken into account by correcting the molecular diffusion coefficient with tortuosity (*AICHE J.* **1958**, 4, 343-345; *J. Hazard. Mater.* **2020**, 384, 121420). As for the bubbles, although they can perturb the boundary layer, the retention of bubbles in the pores also hinders the local mass transfer. In addition, these two issues do not alter the shorter mass transfer distance and the nonuniform surface potential distribution in the small channels, which are two core points in our mechanism. In summary, our main objective was to clearly reveal the differences between various pore sizes. Therefore, it was reasonable to use the simplified microchannel model.

Overall, the good fitting results can be attributed to the rational selection of the parameters. All parameters are within reasonable ranges. By highlighting the suppression of the diffusion layer and the nonuniform distribution of surface potential, this model provides a good explanation for the experimental results. Finally, to enhance the comprehensiveness of this paper, we supplement an analysis of the limitations of the simulation in the discussion section. Accordingly, the data and discussions have been updated as follows. The limitations of our model were also discussed in the Discussion section.

Line 385 – 396 in the revised manuscript:

“... A strong agreement was shown between the simulation results and the experimental data at different current densities, confirming the effectiveness of our model.

Fig. 5 Multiphysics simulation results of the microreactors. ... c Simulated pseudo-first-order kinetic constants of 4-CP degradation. ...”

Line 430 – 432 in the revised manuscript:

“Although factors such as the irregular pore structure and generation of oxygen bubbles were not taken into consideration in the simplified model, its accuracy in elucidating spatially confined oxidation processes is adequate.”

Other relevant statements were added to the Supplementary Information.

Line 119 – 121 in the revised Supplementary Information:

“... When the simulation was performed at different parameter values (e.g., pore size and applied current), all other parameters remained unchanged.”

“

Supplementary Fig. 32. Effluent 4-CP concentration as a function of HRT in different simulated channels. The effluent 4-CP concentration as a function of HRT at current densities of (a) 39.3 mA cm^{-2} , (b) 19.7 mA cm^{-2} , and (c) 9.8 mA cm^{-2} , respectively. The dashed lines represent the fitted degradation curves.”

“Supplementary Table 4. Parameters and values used in the simulation.

Parameters	Value
Oxygen evolution potential ^a	2.1 [V]
Exchange current density of oxygen evolution	4×10^{-2} [A m ⁻²] ²⁵
Electron transferred coefficient for oxygen evolution ^b	0.15
Conductivity of electrolyte	2.54 [S m ⁻¹]
Applied current of one channel ^c	0.059, 0.278, 1.203, 5.333 [μA]
Inlet concentration	1×10^{-4} [M]
HRT for 4-CP simulation	2.7 – 54.5 [s]
HRT for TA simulation	2.7 [s]
$k_{g, \bullet OH}$ ^d	8.91×10^{-6}
$k_{4-CP, \bullet OH}$	7.6×10^6 [m ³ s ⁻¹ mol ⁻¹] ²⁶
$k_{TA, \bullet OH}$	4.4×10^6 [m ³ s ⁻¹ mol ⁻¹] ²²
$k_{4-CP, DET}$ ^e	0.386 [m ³ mol ⁻¹]
$k_{\bullet OH, \bullet OH}$	5.5×10^6 [m ³ s ⁻¹ mol ⁻¹] ²⁷
$k_{4-CP \text{ products}, \bullet OH}$ ^f	5.0×10^6 [m ³ s ⁻¹ mol ⁻¹]
•OH diffusion coefficient	2.2×10^{-9} [m ² s ⁻¹] ⁶
4-CP diffusion coefficient	4.5×10^{-10} [m ² s ⁻¹] ²⁸
TA diffusion coefficient ^g	0.8×10^{-9} [m ² s ⁻¹]
Tortuosity (τ) ^h	1.7 ⁱ

^aThe oxygen evolution potential was determined based on the experiment (Fig. 1c).

^bThe charge transfer coefficient (α) was determined according to other studies. Although a value of $\alpha = 0.5$ is expected on inactive electrodes, several experimental studies have reported $\alpha < 0.5$ for the OER reaction^{2,31}, even as low as 0.10 for Ti₄O₇². A value of $\alpha = 0.15$ is used in the simulation.

^cThe applied current in representative channels was determined according to the geometric area of electrodes (Supplementary equation (5)).

^dThe rate constant for •OH production was determined based on the experiment (Supplementary Fig. 28).

^eThe rate constant for DET reaction of 4-CP was determined based on the parameters optimization in the simulation.

^fThe rate constant for radical oxidation of intermediate product was determined based on previous studies and parameters optimization in the simulation²⁷.

^gThe diffusion coefficient of TA was estimated according to other organic molecules (0.67 – 0.99 m² s⁻¹)³².

^hThe diffusion coefficient was corrected as $\frac{D}{\tau^2}$ ³³, because the tortuosity of the pores may inhibit the diffusion process of molecules³⁴.

ⁱThe tortuosity was estimated according to a previous study (1.2 – 1.8)³⁵.”

REVIEWER COMMENTS

Reviewer #1 (Remarks to the Author):

The revised manuscript of NCOMMS-23-19868A developed a promising model to elucidate the roles of direct electron transfer and radical oxidation for contaminant degradation in spatially confined reactive electrochemical membranes (REM) for water treatment. The authors have made extensive revisions and addressed most of the comments or concerns from the reviewer, especially about the role of surface area increase and diffusion enhancement. This observation will be beneficial for designing future electrochemical systems for water treatment. Hence, this work is recommended for publishing in Nature Communication.

Reviewer #2 (Remarks to the Author):

I have re-reviewed the manuscript corrected by the authors carefully, and the review finds that they have fixed all my concerns well. I think that the revised manuscript can be considered to be published in Nature Communications journal in the present form. I have no other comment.

Reviewer #3 (Remarks to the Author):

Thanks to the authors for responding to my comments, overall I am satisfied, but I still have some questions.

Comments 8, to avoid the effect of hydraulic residence time in single-pass mode, I recommend that the authors treat large volumes of water samples (e.g., 1 L or more) in recirculation filtration mode in related experiments. Then, I believe the authors could get OH radical yields and its contribution to the oxidation of pollutants. You may find else in Journal of Hazardous Materials 2022, 423, 127239.

Comments 10, It is important that the authors provide a modeling approach that can give a general understanding of how pore size affects the oxidation mechanism of pollutants. I also agree with the authors that the pore size distribution and bubbles may not alter the shorter mass transfer distance and the nonuniform surface potential distribution in the small channels in terms of changing trends, but they can significantly affect the accuracy of specific simulation values. The authors defended that it was because of the appropriate selection parameters, however, extremely ideal and simplified models are difficult to give accurate values. For example, as shown in Fig. 4b, the simulation results suggest that the entire 3-mm-thick REM105 possesses OH-producing activity, whereas the most recent experimental results (doi.org/10.1021/acsestengg.3c00181) show that this is not possible. So, the authors need to give a reasonable explanation of how they have managed to match the simulation results so well with the actual measurements.

Point-by-point response to the reviewers' comments

Title: “*Unveiling the spatially confined oxidation processes in reactive electrochemical membranes*”

Manuscript ID: NCOMMS-23-19868A

We sincerely thank all reviewers for their recognition of this work. Their insightful comments are certainly helpful in improving the quality of this manuscript. We have carefully and systematically responded to all the points raised. The reviewers' comments are in bold italic font and our revisions are in blue font. We have also highlighted the revised text in blue in the main text. Provided below are our detailed responses to each point.

REVIEWER #1 (Comments to authors)

General Comment: The revised manuscript of NCOMMS-23-19868A developed a promising model to elucidate the roles of direct electron transfer and radical oxidation for contaminant degradation in spatially confined reactive electrochemical membranes (REM) for water treatment. The authors have made extensive revisions and addressed most of the comments or concerns from the reviewer, especially about the role of surface area increase and diffusion enhancement. This observation will be beneficial for designing future electrochemical systems for water treatment. Hence, this work is recommended for publishing in Nature Communication.

Many thanks to the reviewer for the recognition of this work. Your important review greatly helps to improve the clarity and quality of our manuscript.

REVIEWER #2 (Comments to authors)

General Comment: I have re-reviewed the manuscript corrected by the authors carefully, and the review finds that they have fixed all my concerns well. I think that the revised manuscript can be considered to be published in Nature Communications

journal in the present form. I have no other comment.

We thank the reviewer very much for the time and effort to improve the quality of our manuscript as well as the encouraging comments.

REVIEWER #3 (Comments to authors)

General Comment: Thanks to the authors for responding to my comments, overall I am satisfied, but I still have some questions.

We thank you very much for your recognition and additional comments. Your careful reading and helpful comments will greatly help us improve the quality of our work in the revision. A point-by-point response on all issues raised is as follows.

Comments 8, to avoid the effect of hydraulic residence time in single-pass mode, I recommend that the authors treat large volumes of water samples (e.g., 1 L or more) in recirculation filtration mode in related experiments. Then, I believe the authors could get OH radical yields and its contribution to the oxidation of pollutants. You may find else in Journal of Hazardous Materials 2022, 423, 127239.

We appreciate your important comments. As you mentioned, recirculating and single-pass are two commonly used wastewater treatment systems. We previously fitted degradation curves by the effluent concentration at different HRTs in the single-pass mode, as it is more commonly used in practical wastewater treatment (*J. Hazard. Mater.* **2022**, 423, 127239). Based on your valuable suggestions, we supplemented the •OH production and its contribution to the oxidation of pollutants on REMs in the recirculation mode (Fig. S25 and S26). Considering that the electrode area used in our study ($2.54 \times 10^{-4} \text{ m}^2$) is significantly smaller than that in *Journal of Hazardous Materials 2022, 423, 127239* ($8.48 \times 10^{-3} \text{ m}^2$), we utilized a relatively smaller volume of water sample (i.e., 300 mL) in the recirculation mode. Overall, the results obtained under recirculation mode were in a similar trend to those under single-pass mode. We believe that it is adequate to demonstrate the point in our manuscript.

Using 2-hydroxyterephthalic acid (HTA) as a probe, we investigated the production of •OH at recirculation mode (Fig. S25). As the pore size of REM decreased, the production of HTA gradually decreased, which is consistent with the previous

experimental results. Considering that HTA may also undergoes electrooxidation, we selected the initial growth stage for linear fitting, where the generation process of HTA predominates. Additionally, the degradation of terephthalic acid (TA) also followed a similar trend to the generation of HTA, which is facilitated on REM_{105μm}.

The degradation performance in the recirculation mode on REMs is shown in Fig. S26. The 4-CP degradation performance on REM_{7μm} was also superior to REM_{105μm} under recirculation mode, with a pseudo-first-order kinetic constant 3 times higher than that on REM_{105μm}. This result is consistent with the single-pass mode. The enhancement factor on REM_{7μm} in the recirculation mode was slightly lower than that in the single-pass mode (i.e., 4.1 times), because the mass transfer limitation was alleviated when the reactant was relatively adequate (from 100 μM to 54 μM for REM_{105μm}) under the recirculation mode. The addition of TBA (tert-butyl alcohol) resulted in varying degrees of inhibition for REMs. REM_{7μm} showed a smaller degree of inhibition (13%) compared to REM_{105μm} (43%). The result indicates that radical oxidation plays a more significant role in the REM with a large pore size, which is consistent with the result in the single-pass mode.

In summary, related experiments were supplemented under recirculation mode. The results are generally consistent with those obtained from the single-pass mode, which strengthens our conclusion. The relevant statements and figures have been added to the manuscript and Supplementary Information.

Line 274 – 281 in the revised manuscript:

“In addition, the •OH production and its contribution to electrooxidation were measured in a recirculation mode⁵³. REM_{105μm} demonstrates a superior capability in TA degradation and HTA production (Supplementary Fig. 25), with an HTA generation rate four times higher than REM_{7μm}. Regarding the degradation of 4-CP, the pseudo-first-order kinetic constant on REM_{7μm} is three times higher than that on REM_{105μm}, while the contribution of radical oxidation on REM_{7μm} is much smaller (Supplementary Fig. 26). Overall, the results obtained under the recirculation mode are well consistent with those of the single-pass mode.”

Page 30 – 31 in the revised Supplementary Information:

“

Supplementary Fig. 25. The production of $\bullet\text{OH}$ in a recirculation mode. a-d The decay of TA and production of HTA in the recirculation mode on (a) $\text{REM}_{7\mu\text{m}}$, (b) $\text{REM}_{17\mu\text{m}}$, (c) $\text{REM}_{45\mu\text{m}}$, and (d) $\text{REM}_{105\mu\text{m}}$. The initial concentration of TA: $100\ \mu\text{M}$. Current: $19.7\ \text{mA cm}^{-2}$. Volume of water sample: $300\ \text{mL}$. Electrolyte: $0.33\ \text{M NaClO}_4$. Flux: $0.066\ \text{mL cm}^{-2}\ \text{s}^{-1}$. The dotted line represents the linear regression curve obtained from the data in the initial 30 min. Error bars represent the data from duplicate tests.

Supplementary Fig. 26. The degradation performance of 4-CP in the recirculation mode. a, c, e, g The effluent 4-CP concentration as a function of HRT on **(a)** REM_{7µm}, **(c)** REM_{17µm}, **(e)** REM_{45µm}, and **(g)** REM_{105µm}. **b, d, f, h** The corresponding pseudo-first-order kinetic constants on **(b)** REM_{7µm}, **(d)** REM_{17µm}, **(f)** REM_{45µm}, and **(h)** REM_{105µm}. The initial concentration of 4-CP: 100 µM. Current: 19.7 mA cm⁻². Volume of water sample: 300 mL. Flux: 0.066 mL cm⁻² s⁻¹. Error bars represent the data from duplicate tests.”

Comments 10, It is important that the authors provide a modeling approach that can give a general understanding of how pore size affects the oxidation mechanism of pollutants. I also agree with the authors that the pore size distribution and bubbles may not alter the shorter mass transfer distance and the nonuniform surface potential distribution in the small channels in terms of changing trends, but they can significantly affect the accuracy of specific simulation values. The authors defended that it was because of the appropriate selection parameters, however, extremely ideal and simplified models are difficult to give accurate values. For example, as shown in Fig. 4b, the simulation results suggest that the entire 3-mm-thick REM105 possesses OH-producing activity, whereas the most recent experimental results (doi.org/10.1021/acsestengg.3c00181) show that this is not possible. So, the authors need to give a reasonable explanation of how they have managed to match the simulation results so well with the actual measurements.

We greatly appreciate your comments on our manuscript. The model in our manuscript is highly simplified and comparable to a one-dimensional simulation of the reactive electrochemical membrane. In this case, the pore size and porosity of the REMs were taken into account. The influence of the pore structure on the mass transfer was taken into consideration by correction of tortuosity (*AICHE J.* **1958**, 4, 343-345; *J. Hazard. Mater.* **2020**, 384, 121420). This correction affects the reaction kinetics, and allows us to match the simulated results with the experimental results. It is worth noting that the correction cannot influence the trend, but only affects the values of the kinetic constants. In addition, the correction well falls within a reasonable range (*J. Mater. Res.* **2013**, 28, 2444-2452). To further analyze the impact of tortuosity on the simulated degradation performance, we supplemented the simulations of reaction kinetics at different tortuosity values (Fig. S37a). As the tortuosity decreases, the reaction kinetics of the channels with smaller pore sizes were hardly affected. However, the reaction kinetics of the channels with larger pore sizes decreased as the tortuosity increased. This result aligns well with our experimental results, where the kinetics of the REMs with large pore sizes were controlled by the mass transfer process.

The simulated production of HTA primarily depends on the applied current, the percent yield of HTA, and the potential distribution. Other factors such as the pore structure are less significant, as the highly reactive $\bullet\text{OH}$ will be easily captured by the abundant TA at both the experimental and simulated conditions. In the simulation, the applied current was the same as the value in the experiment. The potential distribution in the porous electrode was calculated by finite element simulation, which has been widely studied in the field of batteries (*Adv. Energy Mater.* **2022**, 12, 2201506). From a macroscopic perspective, it is greatly influenced by the porosity of the electrode and the conductivity of the electrolyte, the values of which were obtained from the

experiment and used in our simulation. Therefore, the discrepancy between the simulation in our manuscript and the experimental research (*ACS EST Engg.* **2023**, 3c00181) is not due to the simplification of our model.

In fact, the potential is also influenced by the properties (e.g., transfer coefficient (α), onset potential (E_o)) and experimental condition (e.g., applied current density) of the electrode (*J. Electroanal. Chem.* **1995**, 397, 27-33). All these parameters we used in the simulation were obtained from literature or experiment. To enhance the rigor in our research, we supplemented the simulation and discussion about these parameter values (Fig. S37).

(1) A relatively low α was used in our simulation (*ACS ES. T. Eng.* **2022**, 2, 713-725), which leads to a more uniform distributed surface potential (*J. Electroanal. Chem.* **1995**, 397, 27-33) (Fig. S37c). Therefore, no significant surface potential drop was observed in the simulation of channel_{105 μ m}.

(2) The overall potential distribution was also influenced by the oxygen evolution capability of the electrode material. A higher E_o led to a higher electrode potential throughout the channels (Fig. S37b).

(3) The uniformity of the potential distribution was also influenced by the applied current (potential) (Fig. S37d). The current was concentrated on the electrode surface when a high current was applied, which is consistent with the published research (*ACS EST Engg.* **2023**, 3c00181). The applied current is relatively low in our work, which alleviates the potential drop in the channel_{105 μ m}.

In addition, a relatively high concentration of electrolyte (330 mM NaClO₄) was used both in the experiment and the simulation, which also expanded the length with •OH-producing activity (*ACS EST Engg.* **2023**, 3c00181). Based on the above reasons, the entire channel_{105 μ m} possesses •OH-producing activity. We believe the simulation results were not contradictory with the published research (*ACS EST Engg.* **2023**, 3c00181). It is worth mentioning that the references of the parameters used in the simulation have been elaborated in the Supplementary Information.

We also tried to visualize the potential distribution in REMs using Pb²⁺ electrodeposition imprinting (Fig. S30), which is a valuable method mentioned in the reference provided (*ACS EST Engg.* **2023**, 3c00181). It can be observed that PbO₂ was deposited in the entire length of REMs. The deposition in REM_{7 μ m} was more nonuniform than in REM_{105 μ m}. This experimental result qualitatively confirms our simulation of the potential distribution.

Overall, the properties of the electrode materials and the operating conditions

influence the surface potential distribution, which in turn determines the length with •OH-producing activity. To further enhance the rigor of our study, we supplemented the discussions on the simulation parameters related to potential and reaction kinetics. Additionally, we validated our simulation by PbO₂ electrodeposition. We believe that these results sufficiently demonstrate the reasonability of our simulation. The relevant statements and figures have been added to the manuscript and Supplementary Information.

Line 320 – 324 in the revised manuscript:

“The potential and current distributions were qualitatively visualized using Pb²⁺ electrodeposition imprinting⁵⁶ (Supplementary Fig. 30). Although PbO₂ can be deposited throughout the entire length of REMs, its distribution in REM_{7μm} is rather nonuniform, indicating the unevenly distributed surface current (Supplementary Fig. 29).”

Line 437 – 442 in the revised manuscript:

“Moreover, the potential distribution is also influenced by the properties (e.g., transfer coefficient, onset potential for oxygen evolution, and tortuosity) and operational parameters (e.g., applied current) of the electrode, as indicated by the simulations (Supplementary Fig. 37). This suggests that reactions within the pores can be precisely controlled by regulating the electrode structure and operational conditions.”

Line 71 – 72 in the Supplementary Information:

“... , which is a significant factor that influence the surface potential distribution in porous electrode³.”

Page 35 in the revised Supplementary Information:

“

Supplementary Fig. 30. Visualization of the potential distribution in REMs. a-d The cross-sectional SEM image, and Pb elemental mapping on (a) REM_{7μm}, (b) REM_{17μm}, (c) REM_{45μm}, and (d) REM_{105μm}. Scale bars: 800 μm. Current: 19.7 mA cm⁻². Reaction time: 60 min. Electrolyte: 0.05 M Pb(NO₃)₂, 0.33 M NaClO₄.”

Page 42 in the revised Supplementary Information:

“

Supplementary Fig. 37. Simulation of the reaction kinetics and potential distribution in the channels. **a** Simulated reaction kinetics at different tortuosity values. **b** Simulated surface potential distribution in channel_{105 μm} and channel_{7 μm} at different onset potentials (E_o). **c** Simulated surface potential distribution in channel_{105 μm} and channel_{7 μm} at different transfer coefficients (α). **d** Simulated surface potential distribution in channel_{105 μm} and channel_{7 μm} at different applied currents. Due to the slow mass transfer in large channels, the reaction kinetics are greatly affected by tortuosity. The region with •OH-producing activity expands with the increase of E_o . A higher applied current and a larger α lead to a more nonuniform potential distribution. The simulated results follow a similar trend to other studies^{25,26}.”